# Transformation Rate Maps of Dissolved Organic Carbon in the Contiguous U.S.

Lingbo Li[1], Hong-Yi Li [1*], Guta Abeshu[2], Jinyun Tang[3], L. Ruby Leung[2], Chang Liao[2], Zeli Tan[2], Hanqin Tian[4], Peter Thornton[5], Xiaojuan Yang[5]

[1]Department of Civil and Environmental Engineering, University of Houston, Texas, USA
[2]Pacific Northwest National Laboratory, Washington, USA
[3]Lawrence Berkeley National Laboratory, California, USA
[4]Boston College, Massachusetts, USA
[5]Environmental Sciences Division, and Climate Change Science Institute, Oak Ridge National Laboratory, Tennessee, USA

*Correspondence to*: Hong-Yi Li (hongyili.jadison@gmail.com)

**Abstract.** Riverine dissolved organic carbon (DOC) plays a vital role in regional and global carbon cycles. However, the processes of DOC conversion from soil organic carbon (SOC) and leaching into rivers are insufficiently understood, inconsistently represented, and poorly parameterized, particularly in land surface and Earth system models. As a first attempt to fill this gap, we propose a generic formula that directly connects SOC concentration with DOC concentration in headwater streams, where a single parameter, the transformation rate from SOC in the soil to DOC leaching flux, $P_r$, accounts for the overall processes governing SOC conversion to DOC and leaching from soils (along with runoff) into headwater streams. We then derive high-resolution $P_r$ maps over the contiguous U.S. (CONUS) using SOC data from two different sources: the Harmonized World Soil Database v1.2 (HWSD) and SoilGrids 2.0. Both maps are developed following the same five major steps: 1) selecting independent catchments where observed riverine DOC data are available with reasonable quality; 2) estimating catchment-average SOC for the independent catchments; 3) estimating the $P_r$ values for these catchments based on the generic formula and catchment-average SOC; 4) developing a predictive model of $P_r$ with machine learning (ML) techniques and catchment-scale climate, hydrology, geology, and other attributes; and 5) deriving a national map of $P_r$, based on the ML model. For evaluation, we compare the DOC concentration derived using the $P_r$ map and the observed DOC concentration values at evaluation catchments. The resulting mean absolute scaled error and coefficient of determination are 0.73 and 0.47 for the HWSD-based model and 0.58 and 0.72 for the SoilGrids-based model, respectively, suggesting the effectiveness of the overall methodology. Efforts to constrain uncertainty and evaluate sensitivity of $P_r$ to different factors are discussed. To illustrate the use of such maps, we derive a riverine DOC concentration reanalysis dataset over CONUS. The two $P_r$ maps, robustly derived and empirically validated, lay a critical cornerstone for better simulating the terrestrial carbon cycle in land surface and Earth system models. Our findings not only set a foundation for improving our predictive understanding of the terrestrial carbon cycle at the regional and global scales, but also hold promises for informing policy decisions related to decarbonization and climate change mitigation.

## 1 Introduction

With the Earth's climate rapidly warming due to increasing atmospheric greenhouse gas concentrations, there is a growing focus on quantifying the regional and global carbon pools within the land, riverine, and oceanic systems, as well as the intricate interconnections among them (Duarte, 2017; Jing et al., 2021; Teodoru et al., 2015). Each year, about 2 billion metric tons of dissolved organic carbon (DOC) are transported from land to the oceans via rivers globally, comparable to the amount of atmospheric $CO_2$ that deposits into the ocean (Hansell et al., 2009; Lønborg et al., 2020). Moreover, riverine DOC is vital to aquatic biogeochemistry by providing nutrients to microbial communities and influencing aquatic greenhouse gas emissions (Li et al., 2019).

However, it remains a challenge to represent and predict riverine DOC effectively in the land biogeochemical module of Earth system models, which are the primary tools for studying carbon cycles in the context of climate change. A chief reason behind this long-standing challenge is the complexity of terrestrial and aquatic processes and their interactions governing SOC transformation to DOC and transport from soils to rivers. The relevant terrestrial processes include the conversion of solid SOC into soil DOC, the adsorption and desorption of DOC by surrounding soils, the transport of DOC from soils into headwater streams along with runoff, and the degradation of soil DOC during this transport. These processes are further influenced by numerous biotic factors, such as microbial, plant, and enzymatic activities, as well as abiotic factors, including soil temperature, moisture, pH (Davidson and Janssens, 2006; Kaiser and Kalbitz, 2012; Kalbitz et al., 2000; Sinsabaugh, 2010). The relevant aquatic processes include the transportation of riverine DOC from headwater streams, the interception of DOC fluxes by reservoirs and lakes, the degradation of riverine DOC during transport, and the consumption of DOC by aquatic biosystems. Furthermore, each process is controlled by several environmental factors, which often exhibit substantial spatial heterogeneity. Models attempt to represent these complexities through parameters associated with governing equations. For instance, Tian et al. (2015a, b) incorporated the effects of runoff on DOC leaching with a coefficient that involves both surface and subsurface runoff. Surface and subsurface runoff are further affected by many environmental factors such as climate, soil, vegetation, and topography (Li et al., 2014; Li and Sivapalan, 2014).

The complexity of relevant processes and their driving environmental factors is also evident in the diverse process descriptions in several land biogeochemical models that are pioneers in representing the suite of processes from SOC to riverine DOC, such as Dynamic Land Ecosystem Model (DLEM) (Tian et al., 2015a, b; Yao et al., 2021), the integrated catchment model for carbon (INCA-C) (Futter et al., 2007), the Joint UK Land Environment Simulator Dissolved Organic Carbon model (JULES-DOCM) (Nakhavali et al., 2018), and the TRIPLEX-hydrological routing algorithm (TRIPLEX-HYDRA) (Li et al., 2019).

These models differ in the processes involved and the process descriptions, owing to the inconsistent understanding of relevant processes among the modeling community. For instance, DLEM and TRIPLEX-HYDRA both adopt CENTURY-like (Metherell et al., 1993; Parton et al., 1987) formulas to estimate DOC leaching fluxes (Tian et al., 2015a, b; Yao et al., 2021; Li et al., 2019), but with notably different ways of incorporating both soil and water-related factors. For instance, TRIPLEX-HYDRA includes an empirical coefficient to account for soil absorption of SOC before its dissolution and DOC degradation

in soils, which are not explicitly accounted for in DLEM. TRIPLEX-HYDRA incorporates hydrologic effects by directly using the water flow rate, whilst DLEM uses a dimensionless ratio to account for these effects. Equally important, the available observations have not been fully used for estimating or calibrating the numerous DOC-related parameters at the regional and larger scales in a spatially continuous yet variable fashion. Existing models usually calibrate several DOC-related parameters against DOC observations at a limited number of river stations, leading to overparameterization, where multiple combinations

of parameter values can achieve the same simulation results (Sivapalan, 2005). Moreover, the resulting parameters often poorly reflect the spatial heterogeneity of underlying processes and environmental factors due to the limited spatial coverage of DOC observations (Futter et al., 2007; Tian et al., 2015a, b; Nakhavali et al., 2018; Li et al., 2019; Liao et al., 2019; Yao et al., 2021). Overall, existing models for simulating DOC fluxes are still subject to limited transferability over poorly observed regions due to insufficient process understanding, data scarcity, and overparameterization.

    One traditional strategy for improving model transferability over poorly observed regions is parameter regionalization. Generally, low-dimensional relationships between a target parameter and other environmental variables are derived based on prior knowledge or regression analysis from the locations where sufficient observations are available. The relationships are then generalized and transferred to poorly-observed places (Alebachew et al., 2014; Ayata et al., 2018; Doron et al., 2011;

Dupas et al., 2013; Tan et al., 2022; Ye et al., 2014). However, such a strategy will not work well if statistically robust and mechanistically meaningful relationships can not be derived from the conventional regression analyses or prior knowledge when, for example, the relationships are high-dimensional and nonlinear (Abeshu et al., 2022; Li et al., 2022). Fortunately, state-of-the-art machine learning (ML) techniques offer a promising and effective alternative strategy, owing to their proven advantages in capturing higher-order relationships between the target and predictive variables, especially when prior

knowledge of such relationships is still in its infancy (Afan et al., 2016). For example, ML techniques have been successfully employed to capture the complex relationships between median sediment particle size and several environmental factors, which enabled the derivation of a national map of median sediment particle size (Abeshu et al., 2022). They have also been used to predict the concentration of fecal indicator bacteria, providing valuable guidance to beach closure problems (Li et al., 2022).

As the first step in addressing these challenges, this study develops an ML-powered approach for parameterizing DOC leaching fluxes at regional and continental scales. The rest of this paper is organized as follows. Section 2 outlines the overall methodology, including governing equations and corresponding parameters, data preparation, and the ML techniques employed. Section 3 presents the results over the contiguous United States (CONUS). Sections 4, 5, and 6 discuss the uncertainty, potential use of the resulting datasets, limitations of methods, and data availability. Section 7 concludes with a summary and potential future directions.

## 2 Methods

The methodology here is described with specific details over the CONUS region, but it is transferable to other regions after some modifications based on data availability.

### 2.1 Governing Equation

Several existing land or land biogeochemical models commonly employ CENTURY-like formulas to represent the leaching of DOC (Futter et al., 2007; Tian et al., 2015a, b; Nakhavali et al., 2018; Li et al., 2019; Yao et al., 2021; Parton et al., 1998). In such formulas, the DOC leaching flux is estimated as a linear function of several factors, including the SOC or DOC concentration in soil, runoff, and other relevant environmental factors. For example, in DLEM (Tian et al., 2015a, b), DOC leaching flux is estimated as

$$F_{DOC\_runoff} = F_{SOC\_Soil} \times \alpha1 \times \alpha2 \times \alpha3 \tag{1}$$

where $F_{SOC\_Soil}$ is the total amount of decomposed SOC in soil (g Cm$^{-2}$s$^{-1}$); $\alpha1$ is the fraction of decomposed SOC that is dissolvable (%); $\alpha2$ is the runoff coefficient (-), i.e., the ratio of total runoff volume to the sum of total runoff volume and soil water content; and $\alpha3$ is another coefficient (-) accounting for the effects of DOC concentration in soil water and desorption. In TRIPLEX-HYDRA (Li et al., 2019), DOC leaching flux is given as

$$F_{DOC\_runoff} = C_{SOC} \times K_s \times K_a \times Q_{runoff} - K_{soil} \tag{2}$$

where $F_{DOC\_runoff}$ is the DOC flux in the soil water (g C/s); $C_{SOC}$ is the concentration of SOC in the soil (g C/m$^3$); $K_s$ is the solubility of SOC (-); $K_a$ is the adsorption coefficient of SOC (-); $K_{soil}$ represents the degradation rate of DOC in soils (g C/s), and $Q_{runoff}$ is total runoff rate (m$^3$/s).

Based on the similarity between equations (1) and (2), while keeping minimal complexity in the process representation, we propose a simpler formula to estimate DOC leaching flux as

$$F_{DOC\_runoff} = C_{SOC} \times Q_{runoff} \times P_r \tag{3}$$

Eqn. (3) can be rewritten as

$$C_{DOC\_runoff} = \frac{F_{DOC_{runoff}}}{Q_{runoff}} = C_{SOC} \times P_r \qquad (4)$$

where $F_{DOC\_runoff}$ is the DOC leaching flux (g C/s), $C_{SOC}$ is the SOC concentration (g C/m³ soil), $Q_{runoff}$ is the runoff volume per unit time (m³ water/s), $P_r$ is the transformation rate from SOC in soil to DOC in runoff (m³ soil/ m³ water), and $C_{DOC\_runoff}$ is the DOC concentration in the runoff (g C/m³ water).

Eqn. (4) has two advantages: 1) its lumped parameter, $P_r$, accounts for all relevant processes and factors, including soil carbon decomposition, DOC sorption-desorption balance, DOC transport and degradation in soils, etc.; 2) its simplicity significantly reduces data requirements for large-scale parameterization since it is highly parameter-parsimonious and much more compatible with the availability of DOC observational data.

For a "small catchment", we further assume that $C_{DOC\_runoff}$ can be approximated with the riverine DOC concentration at the catchment, i.e.

$$C_{DOC\_outlet} \approx C_{DOC\_runoff} \qquad (5)$$

where $C_{DOC\_outlet}$ is the riverine DOC concentration at the catchment outlet (g C/m³). In this study, a "small catchment" refers to the drainage basin extending from the river station upstream to the furthest tributaries that do not have any upstream rivers. Note that a small catchment is not necessarily a headwater catchment that includes only one river (He et al., 2024). The rationale behind Eqn. (5) is two-fold: 1) the travel time of runoff in streams of small catchments is typically much less than one day, e.g., the daily total runoff rate can be approximated with the daily streamflow rate for small catchments (Ducharne et al., 2003; Li et al., 2013), and 2) the degradation rate of DOC in headwater streams is approximately 1% per day, based on our literature review of existing experimental (Qualls and Haines, 1992; Sobczak et al., 2003) and modeling studies (Tian et al., 2015a, b; Li et al., 2019) (for a full list of references, see Supplementary Table S1). Given this minimal degradation rate and the short residence time of DOC in streams of small catchments (on the order of a few hours), it is reasonable to assume negligible DOC degradation from the point it enters the stream to the point it exits into downstream rivers. Combining Eqn. (4) and (5) yields

$$C_{DOC\_outlet} \approx C_{SOC} \times P_r \qquad (6)$$

Eqn. (6) may be used in at least two ways: 1) One can estimate $P_r$ at the catchment scale wherever observed DOC concentration and SOC values are available, and 2) Once $P_r$ is estimated a priori or through calibration, one can predict riverine DOC concentration or discharge in streams of small catchments from the corresponding SOC values.

## 2.2 Data

DOC observations are available via the Water Quality Portal (WQP) (Water Quality Portal, 2021). WQP integrates the publicly available water quality data from the USGS National Water Information System (NWIS) (U.S. Geological Survey), the EPA STOrage and RETrieval Water Quality eXchange (STORET-WQX) (USEPA), and the USDA ARS Sustaining The Earth's

Watersheds - Agricultural Research Database System (STEWARDS) (Steiner et al., 2008). As of now, the WQP features data from 32071 river stations within the CONUS. These stations have recorded at least one DOC measurement between 1900 and the present.

Regional and global soil property maps, such as soil organic carbon (SOC) maps, are typically generated using two primary methods: the linkage method (also known as the taxotransfer rule-based method) (Batjes, 2003) and digital soil mapping (McBratney et al., 2003). This study employs the most widely recognized datasets from each method: the Harmonized World Soil Database (HWSD) v1.2 (Fischer et al., 2008) and SoilGrids 2.0 (Poggio et al., 2021). HWSD provides SOC data at a spatial resolution of 1 km for two soil layers—the top layer (0–30 cm) and the sub-layer (30–100 cm). As one of the first

globally harmonized soil datasets, it integrates data from diverse national and regional sources into a standardized framework, making it a foundational resource for many Earth system modeling studies (Best et al., 2011; Han et al., 2014; Todd-Brown et al., 2013; Zhao et al., 2018). SoilGrids 2.0 offers SOC data at a higher resolution of 250 m for the same layers, leveraging machine learning algorithms to enhance accuracy and constrain uncertainty. Its higher resolution and improved reliability have made it increasingly popular for Earth system modeling since its release (Dai et al., 2019; Hengl et al., 2017; Poggio et al.,

2021). Considering that DOC leaching from soils into rivers predominantly comes from the topsoil (Brooks et al., 1999; Finlay et al., 2006), we use the SOC content data from the top 30 cm layer for our estimations. We also take into consideration that there are missing values in some grid cells in the HWSD v1.2 and SoilGrids 2.0 and adjust our catchment selection accordingly.

In order to pair up SOC and DOC data at small catchments, we rely on the National Hydrography Dataset Plus (NHDPlus) dataset hosted by the U.S. Geological Survey (USGS) (Mckay et al., 2012). This dataset is chosen for two reasons: Firstly, NHDPlus provides well-defined catchment boundaries and associated river segments, referred to as local catchments and flowlines. It includes ~2.6 million flowlines across CONUS, each linked to a corresponding local catchment that collects lateral runoff into that flowline. Additionally, the upstream drainage catchment for any flowline, which is the sum of both local

catchment and the drainage areas corresponding to all the flowlines upstream of the local one, can be derived from the established flowline network. The sizes of these 2.6 million local catchments vary from the 5th percentile at 0.02 km$^2$ to the 95th percentile at 9.68 km$^2$, depending on the corresponding surface topography, with a CONUS average of 3.12 km$^2$ (Supplementary Fig. S1). Secondly, NHDPlus is closely linked to ScienceBase (Wieczorek et al., 2018), a comprehensive scientific data and information management platform also hosted by USGS. ScienceBase includes a wide range of

environmental variables across 11 categories, such as climate, hydrology, soil, and geological data, conveniently available at the catchment scale across the entire CONUS. These environmental data are critical in the ML modeling analysis.

Correspondingly, the overall data preparation procedure consists of three major steps: 1) Selection of small catchments based on the availability of observed riverine DOC concentrations of adequate quality. 2) Estimation of $P_r$ values for the catchments

selected in Step 1, leveraging the corresponding riverine DOC observations and SOC reanalysis data. 3) Extraction of catchment-scale environmental variables that could potentially influence $P_r$. Specific details of each step will be further discussed in the following subsections. This study adopts two SOC datasets, both of which directly influence the calculated $P_r$ values used in training, thereby affecting all steps leading to the final $P_r$ map. To enhance clarity and avoid redundancy, the HWSD-based model is the primary focus of discussion, as the workflow and major conclusions remain consistent. More information on the SoilGrids-based model is available in the supplementary materials. Users can choose their preferred $P_r$ map based on their specific needs.

### 2.2.1 Selecting small catchments

Our selection process for suimall catchments involves the integration of the NHDPlus dataset and observed riverine DOC concentration data from river stations:

1. We conduct a geospatial analysis to identify the upstream drainage area of each WQP river station using NHDPlus local catchments and flowlines. Using the Python package HyRiver (Chegini et al., 2021), we co-located 29,320 WQP stations with the closest corresponding NHDPlus flowlines. However, 2,751 stations can not be linked due to the absence of adjacent flowlines. When WQP stations are in close proximity and share the same NHDPlus flowline, we retain only the station with the best data availability. For a given flowline, HyRiver traces it back to every upstream flowline, accessing and merging the boundaries of all related NHDPlus local catchments from the Hydro Network-Linked Data Index web server. It also requests the server to simplify the boundaries and split them precisely at the station locations. The relationship between the derived small catchment boundaries and the NHDPlus local catchments is shown in Supplementary Fig. S2a. Through this comprehensive geospatial analysis, we identify the upstream boundaries for 22,201 WQP stations.

2. We further select the WQP stations whose drainage areas can be considered small catchments, based on two criteria: 1) there are no upstream rivers flowing into them, and 2) their drainage areas are no more than 2500 km$^2$. This size threshold ensures that the travel distance of river water (and consequently, DOC) is ~50 km within these catchments. Assuming an average channel velocity of ~1.0 m/s (Chow et al., 1988), the average travel time is ~14 hours, i.e., less than one day. Using these criteria, we identify 18,612 pairs of WQP stations and small catchments.

3. For the 18,612 WQP stations, we perform a rigorous DOC data quality control based on five criteria: a) The record lengths of riverine DOC data should span at least one year; b) There should be at least two riverine DOC observations; c) No single season should dominate the riverine DOC observations, i.e., a single season should not account for more than 50% of the records; d) within the boundaries of the corresponding catchments, there should be sufficient availability of the NHDPlus catchment attributes and SOC reanalysis data; e) the catchments should not be significantly affected by dams, i.e., the total drainage areas of the dams within a catchment should be no more than 5% of the total catchment area. The adoption of criteria (a)-(e) reflects a careful balance between ensuring data quality

and maintaining adequate quantity, ensuring that sufficient WQP stations are retained to represent the entire CONUS. After the data quality control, there remain 5805 WQP stations with their corresponding small catchments.

4.  For the 5805 WQP stations and their small catchments, we verify the spatial independence among them. A catchment is considered nested within another if it lies entirely within the latter's drainage area. While the flux at the downstream catchment's outlet depends on contributions from upstream catchments, the upstream catchments maintain their hydrological independence. As illustrated in Supplementary Fig. S2b, a simple nesting scenario shows two gray catchments, A and B, both located within the red catchment, C. Since A and B have no containing relationship and are both smaller than C, they are classified as independent catchments. In contrast, C is considered a nesting catchment. The same logic applies consistently in more complex nesting scenarios. From the 5805 pairs of the WQP stations and catchments, we identify 2595 as independent and suitable for further ML model training. The other 3210 pairs, despite the nesting issue, are still valuable; they are thus kept for evaluation of estimated DOC (see Sect. 3.4). Due to missing values in SoilGrids 2.0, valid $P_r$ estimates are unavailable for 12 out of 2595 independent catchments; however, the number of evaluation catchments remains unchanged.

### a) $P_r$ of independent catchments

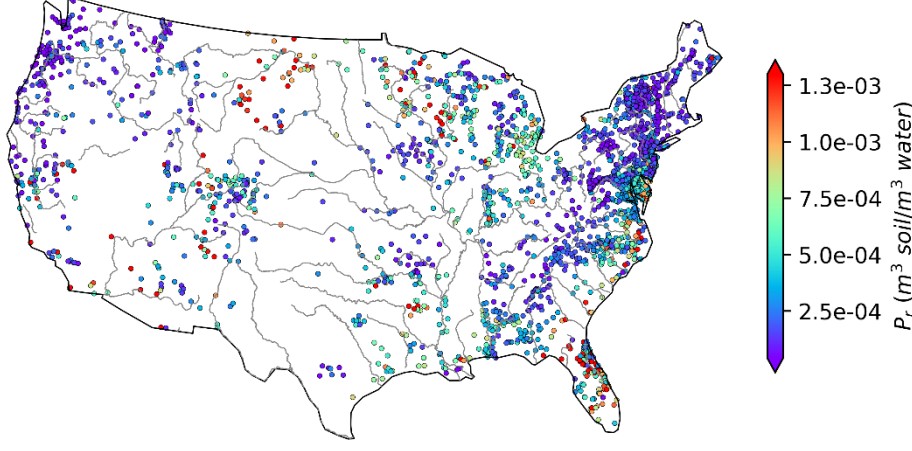

### b) $P_r$ of evaluation catchments

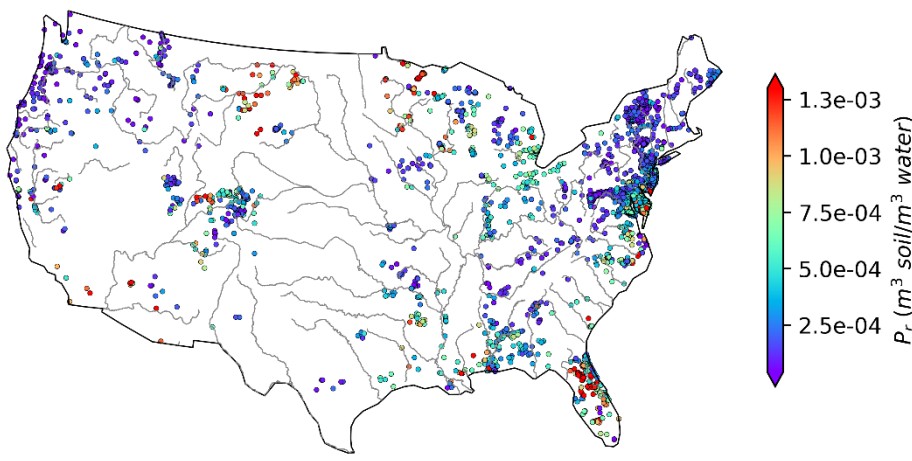

**Figure 1. Variability in estimated $P_r$ across CONUS: a) For independent catchments (n=2595), and b) For evaluation catchments (n=3210). The points indicate the locations of the WQP stations, which are also the outlets of the corresponding small catchments. The CONUS boundary and river shapefiles are directly obtained from open-source datasets GeoPandas (geopandas.org) and Natural Earth (Made with Natural Earth. Free vector and raster map data @ naturalearthdata.com), respectively. The color bars**
**have been adjusted to enhance visual display by showing only the main body of values (from the 5th percentile to the 95th percentile).**

### 2.2.2 Estimating $P_r$

For the final set of the paired WQP stations and small catchments, we calculate $P_r$ using the DOC observation from the WQP stations and long-term mean SOC from HWSD based on Eqn. (6). For each catchment, the catchment polygons are used to
clip the top-layer SOC map at the 1km resolution, and the catchment-scale SOC is subsequently calculated as the spatial

average of SOC values at those 1km grid cells within the catchment. Hereafter the $P_r$ estimated using Eqn. (6) are referred to as "*Estimated $P_r$*". The *Estimated $P_r$*, derived from the analysis of WQP DOC observations and HWSD SOC data, exhibits a wide range of values spanning several orders of magnitude. Figure 1a illustrates the spatial distribution of $P_r$ for the 2595 independent catchments. In these catchments, the Estimated $P_r$ ranges from $4.61 \times 10^{-6}$ to $8.04 \times 10^{-3}$ (m$^3$ soil/ m$^3$ water), with a median value of $2.50 \times 10^{-4}$ (m$^3$ soil/ m$^3$ water). As a broad assessment of the similarity between the catchments used to construct the model and the evaluation catchments, the values of $P_r$ for the evaluation catchments calculated from data values of DOC and SOC using Eqn. (6) are shown in Fig. 1b. Here, the *Estimated $P_r$* values in these catchments range from $8.81 \times 10^{-6}$ to $6.37 \times 10^{-3}$ (m$^3$ soil/ m$^3$ water), with a median of $2.60 \times 10^{-4}$ (m$^3$ soil/ m$^3$ water). Note that the spatial distribution of the selected catchments is quite consistent with the spatial distribution of the WQP stations, i.e., more densely distributed in the eastern than western U.S., suggesting a good spatial representation of the selected catchments over all the WQP stations in CONUS. Figure S8 shows the spatial distribution of *Estimated $P_r$* values derived from the SoilGrids-based model for independent and evaluation catchments. The overall pattern closely resembles that derived from the HWSD-based model. The *Estimated $P_r$* values have a slightly narrower range, from $1.16 \times 10^{-5}$ to $8.69 \times 10^{-3}$ (m$^3$ soil/ m$^3$ water) at independent catchments, and a similar range, from $7.78 \times 10^{-6}$ to $7.55 \times 10^{-3}$ (m$^3$ soil/ m$^3$ water) at evaluation catchments.

### 2.2.3 Extracting environmental variables

We collect 126 environmental variables from the ScienceBase dataset, spanning 11 distinct categories. Seven attributes related to dams and streams are excluded as irrelevant to our objectives, along with 24 attributes containing predominantly zero values (>80%) across CONUS. Of the remaining 95 variables, 46 are relatively independent while 49 showed strong correlations with one or more variables. Following Schober et al. (2018), we define strong correlation as a Pearson correlation coefficient $|r| \geq$ 0.8. The 49 correlated variables are categorized into 9 distinct "correlated groups" based on shared properties, where each variable demonstrates a strong correlation with at least one other variable within its group but a weak correlation ($|r| < 0.8$) with variables outside the group. We address the interdependence within each correlated group through two steps: 1) normalizing individual variables using the Yeo-Johnson power transformation (Yeo and Johnson, 2000) to achieve zero mean and unit variance (Supplementary Fig. S3), and 2) merging the normalized variables through linear summation to create a single new variable (Daoud, 2018). This new variable is now relatively independent of the other environmental variables. For those 46 variables, we apply the same transformation to minimize the impacts of varying magnitudes between different variables. Eventually, 54 variables remain, including 46 originally relatively independent and 9 newly merged variables from the correlation groups (see Supplementary Tables S2 and S3 for details).

### 2.3 Machine learning techniques

We use the eXtreme Gradient Boosting (XGBoost), which is a powerful and widely adopted ML algorithm due to its exceptional performance in various applications (Abeshu et al., 2022; Delavar et al., 2019; Li et al., 2022). XGBoost is a

scalable end-to-end tree-boosting system that belongs to the ensemble learning family(Chen and Guestrin, 2016). It combines multiple weak learners into a strong learner via sequential training and improving, and eventually forms a robust and accurate predictive model. By using XGBoost in this study, we aim to develop a predictive model that establishes causal linkages
between the target variable, $P_r$, and a small number of environmental variables (denoted as predictors hereafter).

In addition to XGBoost, we take advantage of some other ML tools and techniques. Specifically, we use the Optuna optimization framework (Akiba et al., 2019) and k-fold cross-validation (k=5) for tuning the hyperparameters. By leveraging Optuna and k-fold cross-validation, we can systematically search and optimize the hyperparameters, maximizing the model's performance and accuracy. Furthermore, we employ the SHapley Additive exPlanations (SHAP) (Lundberg and Lee, 2017) to
aid in the selection of environmental factors that are related to $P_r$. SHAP is a technique that assigns importance values to individual predictors in a model, providing insights into their contributions to the prediction. By using SHAP, we can identify the key environmental factors that significantly influence $P_r$ and further refine our model. These techniques have been successfully applied in various studies, including riverine sediment, beach water quality, oceanic particulate organic carbon,
and eutrophication impacts from corn production (Abeshu et al., 2022; Fan et al., 2021; Li et al., 2022; Liu et al., 2021; Romeiko et al., 2020), demonstrating their efficiency and effectiveness in capturing high-dimensional and complex relationships between a target biogeochemical variable and various environmental predictors. Readers are referred to Abeshu et al., (2022) for more details about these techniques.

The overall procedure for developing a predictive ML model is illustrated in Fig. 2 (identical for the SoilGrids-based model) and outlined as follows:

1. Prepare the input data for the ML modelling based on the independent catchments, their corresponding $P_r$ estimates, and environmental variables. To address the substantial statistical disparities and wide variation within each predictor, we employ power transformation on all predictors. The lambda parameter is held constant during the transformation
process for the training, testing, and prediction datasets to ensure consistent and reproducible results. Following the transformation, the dataset exhibits a zero-mean and unit variance, with a distribution that closely resembles a Gaussian distribution (Supplementary Fig. S3).

2. Randomly split the observational dataset (2595 catchments) into two sets: 70% for training and 30% for testing the ML model. These training and testing sets will be used throughout the subsequent steps.
3. Identify the list of predictors out of the 54 environmental variables extracted in Sect. 2.2.3 in three sub-steps:
   a. Generate a completely random predictor.
   b. Prepare an initial list of candidate predictors consisting of the random predictor and an initial list of candidate environmental variables. Use Optuna and k-fold cross-validation to obtain the optimal hyperparameters and train an intermediate ML model until the model achieves the best performance evaluated using the testing
set.

      c.    Calculate and rank the SHAP values for all the candidate predictors. Update the list of candidate predictors by keeping only those predictors with better SHAP values than the random predictor. For example, if the random predictor is ranked 20th, only the top 19 predictors are passed to the next iteration.

      d.    Obtain an almost-final list of predictors by repeating sub-steps b-c.

4. Check the representativeness of the almost-final list of predictors identified in Step 3. For each of these predictors, check whether its values from the independent catchments are statistically representative of the whole CONUS, i.e., its values from those 2.6 million local catchments. Drop those predictors that cannot pass the representativeness check. Similar to Abeshu et al. (2022), the representativeness check on each of the almost-final predictors is performed by comparing the cumulative distribution function (CDF) derived from the observational dataset (2595 training catchments) and the CDF derived from the whole CONUS (about 2.6 million local catchments in NHDPlus). Specifically, comparisons are made between the 5th, 25th, 50th, 75th, and 95th percentiles between the two CDFs. After Step 4, a final list of predictors is obtained.

5. Develop the final ML model based on the final list of predictors using Optuna and k-fold cross-validation methods.

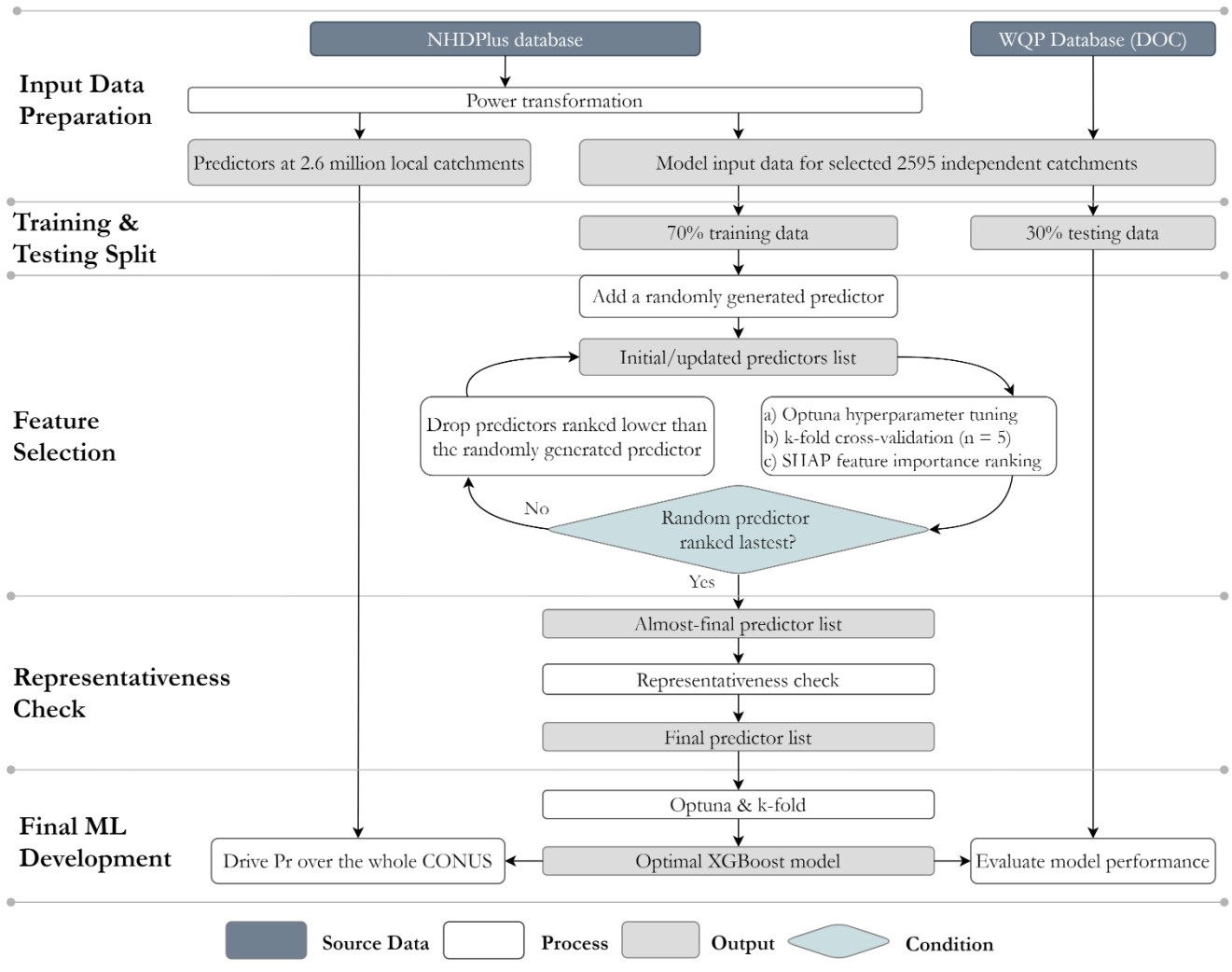

**Figure 2. A workflow for the XGBoost model.**

In Steps 3 and 5, model performance metrics are required for model training and evaluation. The Kling-Gupta efficiency (KGE) (Gupta et al., 2009) has the advantage of simultaneously capturing both the magnitude and phase differences between the observed and simulated series (Abeshu et al., 2022; Gupta et al., 2009). However, further investigations have revealed several
limitations: a) lack of an inherent benchmark value to distinguish between "good" and "bad" model performance, b) sensitivity to outliers, which can result in a systematic overestimation of the target variable, and c) instability when the target variable approaches zero (Knoben et al., 2019; Pool et al., 2018; Santos et al., 2018). Therefore, in addition to KGE, the mean absolute scaled error (MASE) is also used here to alleviate the influence of extreme values in the observation or simulation data
(Hyndman and Koehler, 2006). MASE is a scaled error metric that is defined as the mean absolute error (MAE) of the model simulation divided by scaling factors (MAE of the observation in the original definition). In this study, we normalize MAE by

the geometric mean of the observation data. Note that Steps 3 and 5 above are relatively independent of each other and do not have to rely on the same metrics.

## 3 Results

### 3.1 Predictor selection

In the predictor selection stage, after six iterations of hyperparameter tuning and predictor reduction with KGE as the metric, a list of 15 predictors is selected (blue bars in Fig. 3), including those related to climate, hydrology, pedology, and land cover. In addition, using MASE as the metric in this stage leads to a list of 19 remaining predictors, among which 13 are the same as the list of predictors identified using KGE. The predictor list selected using KGE is preferred due to the fewer predictors and similar model performance. Figure S9 shows the feature selection results (blue bars) for the SoilGrids-based model, with 11 out of 13 predictors also included in the final list derived from the HWSD-based model. This overlap further reinforces the consistency of important features across datasets and enhance the robustness of the selection process.

To enhance the model transferability, we implement a representativeness check (detailed in Sect. 4.1.2) that led to the exclusion of 3 initially selected predictors: "BASIN_AREA," "NLCD01_52," and "NLCD01_95." These variables demonstrated insufficient representativeness of the anticipated real-world data distribution in the prediction phase, resulting in a final model with 12 predictors. Figure 3 presents a comparative analysis of mean absolute SHAP values between the original 15-predictor model (blue bars) and the final 12-predictor model (orange bars). Notably, both models identified the same five dominant predictors, ranked according to their influence in the 12-predictor model: 1) the merged predictor of hydrologic variables ("hydro_related"), 2) the areal percentage of Hydrologic Group BD soil ("HGBD"; detailed classification in Ross et al., 2018), 3) the areal percentage of woody wetlands ("NLCD01_90"), 4) the consecutive wet days ("CWD"), and 5) the subsurface flow contact time ("CONTACT"). The "hydro_related" and "CWD" reflect the overall hydrology condition of a catchment, including runoff, precipitation, and groundwater recharge. Groundwater has a dilution effect on DOC concentration (Kortelainen and Karhu, 2006). Similarly, precipitation and runoff contribute to the distribution and concentration of DOC (Baum et al., 2007; Tranvik and Jansson, 2002; Wilson et al., 2013). Soil type plays a crucial role in determining the soil organic matter quantity and the partitioning of precipitation into runoff, consequently influencing the concentration of DOC in rivers (Autio et al., 2016; Camino-Serrano et al., 2014). Woody wetland, as one land cover attribute, has been identified as a significant predictor of downstream DOC concentration (Duan et al., 2017), because of the enhanced breakdown of organic matter and plant respiration. The influence of subsurface flow contact time on DOC concentration is complex and indirect. For instance, during transport, a catchment with a shorter contact time experiences reduced mineralization loss (Ludwig et al., 1996) and microbial consumption (Helton et al., 2015). Conversely, studies have shown that labile DOC concentration increases with contact time in some alluvial aquifers, as deeper groundwater inflow could provide considerable labile DOC (Helton et al., 2015; Wickland et al., 2012).

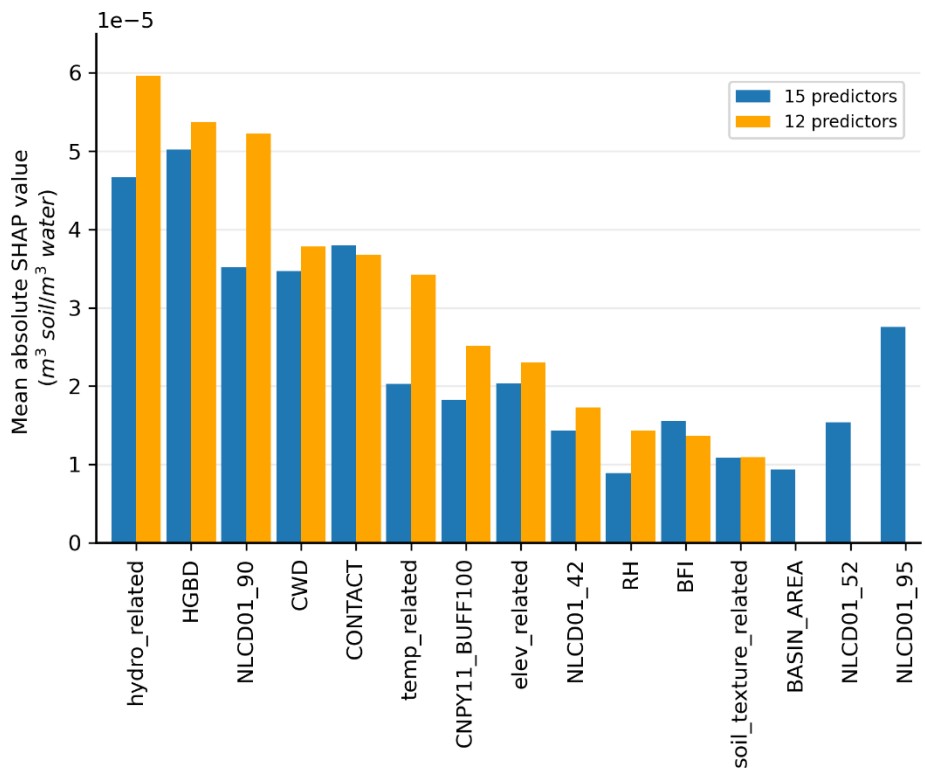

**Figure 3. Mean absolute SHAP values of predictors in models with 15 predictors (blue) and 12 predictors (orange). Note that the SHAP values have the same units as the target variable, $P_r$. Abbreviations: hydro_related (merged predictor representing recharge, runoff, and precipitation); HGBD (areal percentage of Hydrologic Group BD soil); NLCD01_90 (areal percentage of woody wetlands); CWD (consecutive wet days); CONTACT (subsurface contact time); temp_related (merged predictor encompassing**
**potential evapotranspiration, first/last freeze timing, snow fraction, actual evapotranspiration, and mean/min/max temperature); CNPY11_BUFF100 (areal percentage of canopy in the riparian buffer); elev_related (merged predictor for mean/min/max elevation); NLCD01_42 (areal percentage of evergreen forest); RH (relative humidity); BFI (base flow index); soil_texture_related (merged predictor for silt and sand content); BASIN_AREA (catchment area); NLCD01_52 (areal percentage of shrub); NLCD01_95 (areal percentage of herbaceous wetlands). For detailed descriptions, refer to Supplementary Tables S2 and S3.**

## 3.2 Final model

Figure 4 presents the performance of the ML model during both the training and testing phases (phases shown in Fig. 2). To mitigate over-plotting, all the scatter plots (Fig. 4 and hereinafter) employ color coding based on estimated density using kernel density estimation (KDE), as indicated by the corresponding color bar. After the exclusion of the three variables that displayed
poor representativeness, the ML model performance remains stable between the training and testing phases, as gauged by metrics such as MASE, coefficient of determination ($R^2$), and normalized root-mean-square-error (NRMSE). The similarities in these metrics between the *Estimated* and predicted $P_r$ values across both phases support the robustness of our 12-predictor model. Consequently, the final ML model and the subsequent analyses are based on the 12 selected predictors. Furthermore,

the consistency of model performance between the training (MASE= 0.40) and testing (MASE= 0.81) phases suggests that the model overfitting issues are well-regulated (Ying, 2019). We also use KGE as the metric during the final model training. After a comparison between the modeling results using MASE (Fig. 4) and KGE (Supplementary Fig. S4), MASE is preferred for two reasons: a) using MASE yields a better consistency in model performance between the training and testing phases, suggesting better model transferability; b) using MASE leads to a closer agreement between the model simulated and *Estimated* $P_r$ values. Figure S10 illustrates the performance of the SoilGrids-based model, showing similar metrics overall. However, during the testing phase (Supplementary Fig. S10b), the model slightly overestimates low values and underestimates high values. This discrepancy is likely due to the flatter data distribution in the testing dataset, which results in insufficient learning for those extreme values.

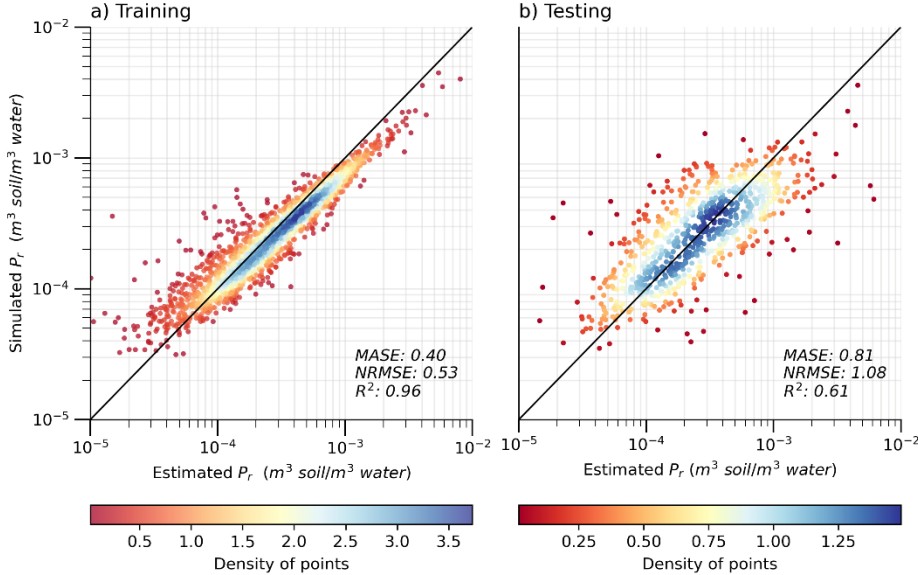

**Figure 4. Performance of the XGBoost model with 12 predictors during a) the training phase (n=1816) and b) the testing phase (n=779). The solid black line indicates a 1:1 ratio. The varying colours indicate the density of points in the scatter plot.**

Table 1 lists the optimized hyperparameter values of the final XGBoost model (Supplementary Table S4 for that of SoilGrids-based model). We choose to tune 8 model parameters, which are critical to the XGBoost tree booster controlling regularization, subsampling, learning process, and the growth of the tree. The optimal values of model hyperparameters are quite different from the default ones, suggesting hyperparameter tuning is necessary.

**Table 1. The optimal values of the XGBoost model hyperparameters.**

| Hyperparameter | Optimal Value | Tuning Range | Default value | Description |
| --- | --- | --- | --- | --- |

| | | | | |
|---|---|---|---|---|
| lambda | $6.725 \times 10^{-1}$ | $[0, \infty]$ | 1 | Control L1 and L2 regularization; the larger the value, the more conservative the model will be |
| alpha | $7.484 \times 10^{-2}$ | $[0, \infty]$ | 0 | |
| gamma | $1.316 \times 10^{-2}$ | $[0, \infty]$ | 0 | Govern the model learning process by changing the step size shrinkage and minimum loss reduction; the larger the value, the more conservative the model will be |
| eta | $1.277 \times 10^{-1}$ | $(0, 1]$ | 0.3 | |
| colsample_bytree | $9.323 \times 10^{-1}$ | $(0, 1]$ | 1 | Control the subsample ratio of columns and training instances; a proper set of those values will prevent the model from over-fitting |
| subsample | $6.142 \times 10^{-1}$ | $(0, 1]$ | 1 | |
| min_child_weight | $8.410 \times 10^{-2}$ | $[0, \infty]$ | 1 | Determine the growth of the tree |
| max_depth | 12 | $[0, \infty]$ | 6 | |

Figure 5 depicts the correlation between $P_r$ and the 12 predictors and among the predictors themselves (Supplementary Fig. S11 for that of SoilGrids-based model), where highly positive correlated and negative correlated are shown in dark-red and blue colors, respectively. Since we have treated the highly correlated variables, the highest positive correlation coefficient is 0.63 between "CNPY11_BUFF100" and "hydro_related", lower than the threshold of 0.8 we adopt in Sect. 2.2.3. Among the observed correlation coefficients, the highest negative correlation coefficient, -0.69, is found between the variables

"elev_related" and "temp_related." This strong negative correlation makes intuitive sense since air temperature decreases with increasing elevation. Note that all of the 12 selected predictors show weak or even negligible correlation with the target variable $P_r$, with the absolute values of the correlation coefficient less than 0.3. It is not surprising since the high-order, nonlinear relations between $P_r$ and the predictors, and likely among the predictors themselves, can only be effectively captured by the ML techniques but not the traditional regression analysis methods.

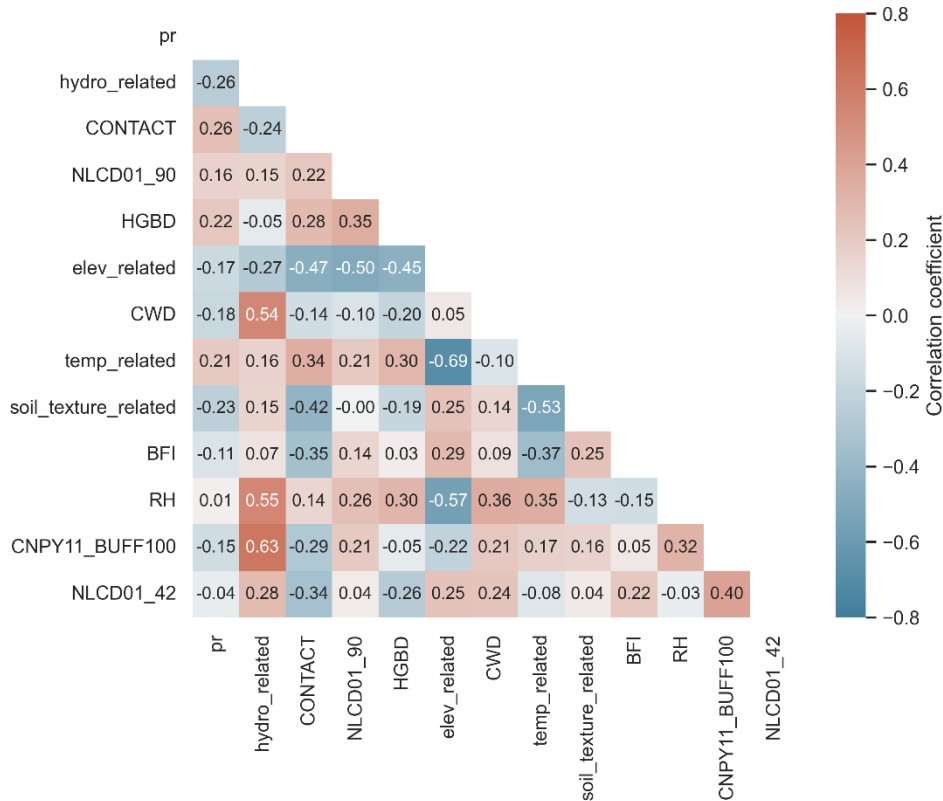

**Figure 5. Covariance heatmap of $P_r$ and the 12 selected NHDPlus predictors. The Pearson correlation coefficient is used. Abbreviations: hydro_related (merged predictor representing recharge, runoff, and precipitation); CONTACT (subsurface contact time); NLCD01_90 (areal percentage of woody wetlands); HGBD (areal percentage of Hydrologic Group BD soil); elev_related (merged predictor for mean/min/max elevation); CWD (consecutive wet days); temp_related (merged predictor encompassing potential evapotranspiration, first/last freeze timing, snow fraction, actual evapotranspiration, and mean/min/max temperature); soil_texture_related (merged predictor for silt and sand content); BFI (base flow index); RH (relative humidity); CNPY11_BUFF100 (areal percentage of canopy in the riparian buffer); NLCD01_42 (areal percentage of evergreen forest). For detailed descriptions, refer to Supplementary Tables S2 and S3.**

### 3.3 $P_r$ map

We develop a spatially continuous map of $P_r$ over CONUS by applying the final XGBoost model over the 2.6 million NHDPlus local catchments, as shown in Fig. 6. The spatial patterns of $P_r$ are generally consistent with those in Fig. 1. High $P_r$ values, shown in orange and red, are mostly located on the southeast coasts, New Mexico, Arizona, southern California, and North Dakota. Low $P_r$ values, shown in blue and purple, are more prevalent in the Northeast and Northwest regions. This consistency between Fig. 1 and Fig. 6 again confirms that the 2595 independent catchments used in the ML modeling are representative of the whole CONUS domain, hence supporting the transferability of the ML modeling results. Figure S12 presents the spatial maps derived using the SoilGrids-based model. The overall patterns are very similar at most places; however, the model

predicts lower values in southern California, New Mexico, and Colorado, and higher values in northern Minnesota and southern
Florida.

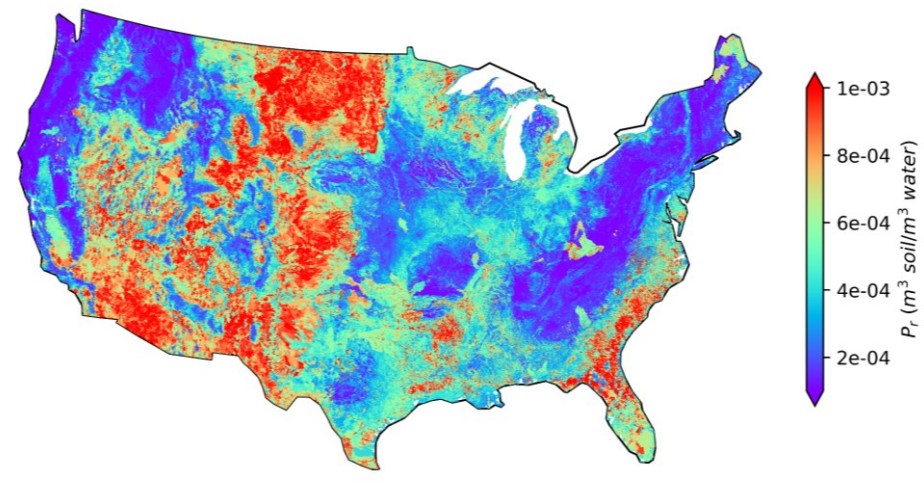

**Figure 6. ML model simulated $P_r$ at over 2.6 million NHDPlus local catchments.**

### 3.4 Evaluation

We evaluate the $P_r$ map by comparing the DOC concentration values derived from this map (and Eqn. 6) with those observed, since there is no direct measurement of $P_r$. The 3210 evaluation stations and their corresponding small catchments (Fig. 1b) are used for this purpose. Note that each of these 3210 evaluation catchments may encompass multiple NHDPlus local catchments. The evaluation thus takes two steps: 1) For each NHDPlus local catchment, calculate its DOC concentration using the predicted $P_r$ value, SOC, and Eqn. (6); derived the DOC concentration for the evaluation catchment (whose outlet is an
observational station) by taking the area-weighted average of local DOC values from the few NHDPlus local catchments located within this catchment; 3) Compare the "derived" DOC concentration with the observed value at the same evaluation catchment. Note that two evaluation catchments are dropped during Step (1) for containing some NHDPlus local catchments without an effective model simulated $P_r$.

Figure 7 shows that our derived DOC concentration values effectively reproduce the spatial variability in the observed values. The MASE, NRMSE and $R^2$ values are 0.73, 1.81, and 0.47, respectively, further suggesting a satisfactory performance. The scattering only occurs to a small portion of the dots, as indicated by the reddish colours. This scattering may stem from several causes, such as the limited availability of DOC observation data and the uncertainties in model development (see Sect. 4 for more details). Despite the scattering, the overall alignment between observed and predicted values suggests that our methods,
including the generic formula and ML modelling, are appropriate and effective. The DOC evaluation performance of the

SoilGrids-based model (Supplementary Fig. S13) reveals a larger systematic bias. This issue is also primarily attributed to differences in data distribution, as the $P_r$ values in evaluation exhibit a wider range than those in training, particularly at low values (see Sect. 2.2.2). Consequently, the model struggles to predict extreme values accurately. For example, for very small $P_r$ values in the evaluation catchments, the model tends to slightly overpredict due to the absence of such small values in the training dataset. Additionally, the typically higher SOC values in these regions further amplify the discrepancies.

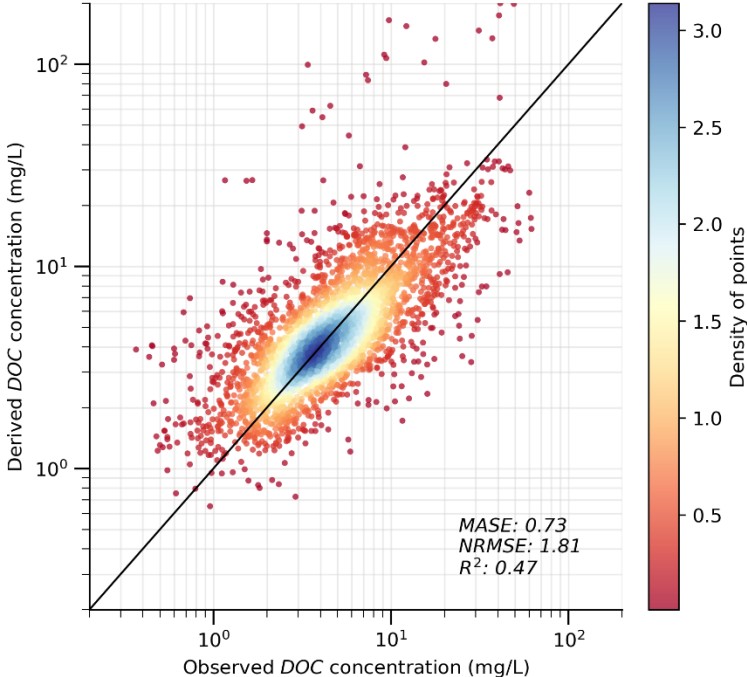

**Figure 7. Evaluation of derived DOC concentration at the catchment scale (n=3208). The solid black line indicates a 1:1 ratio. The varying colours indicate the density of points in the scatter plot.**

## 4 Uncertainty analyses

The final product, our $P_r$ map, is subject to uncertainties from various sources. In this study, we have implemented several measures to constrain the uncertainties embedded in the input data and ML modeling exercise. We also look into the ML model parameter uncertainty via sensitivity analyses.

## 4.1 Efforts to constrain uncertainty

### 4.1.1 Machine learning model input data

The estimation of the DOC long-term average transformation rate, $P_r$, relies on SOC data from the HWSD v1.2 and SoilGrids 2.0 dataset and DOC data from the WQP stations. Despite implementing stringent catchment selection (see Sect. 2.2.1), the challenge of balancing data quantity and quality persists due to limited DOC measurements. Larger uncertainties in $P_r$ are anticipated in catchments with fewer samples or those where most samples are collected in a single season. Additionally, potential uncertainties in the $P_r$ estimation may arise from the mismatch in sampling periods between SOC and DOC datasets. It is crucial to recognize and account for these uncertainties when interpreting and using the $P_r$ map.

The flowline and catchment attributes from NHDPlus constitute the primary inputs in both training and prediction phases for the ML model, and thus may contribute to the uncertainty in the results. NHDPlus catchment attributes are drawn from diverse sources, including remote sensing data and model simulations. Upstream-accumulated values are derived based on flowline data (Wieczorek et al., 2018). A majority of attributes have been compared to equivalent variables, when available, in the Geospatial Attributes of Gages for Evaluating Streamflow version II (GAGESII) dataset (Falcone et al., 2010). These comparisons have demonstrated reasonably strong alignment. Inherent uncertainties may still arise from inaccurate flowline and catchment delineation, inaccuracies in the source data, the conversion of data formats (e.g., from grid-based to catchment-based), and so on. Furthermore, instances of missing data or attributes with zero-inflated values (e.g., regions highlighted in white in Supplementary Fig. S5b) from the NHDPlus dataset can complicate accurate data interpolation by the ML model. Despite the use of the sparsity-aware technique within the XGBoost algorithm, adept at handling missing or zero-inflated data to a certain extent (Chen and Guestrin, 2016), the presence of such challenges persists. Overcoming these limitations is beyond this study's scope.

### 4.1.2 Machine learning model development

In contrast to physical-based models with clearly pre-defined structures, ML models endeavor to discern the optimal structure from input data through the training process. Consequently, uncertainty may emerge at any stage of model development, as detailed in Sect. 2.3. To mitigate model uncertainty, we employ well-established strategies prevalent in diverse applications (Abeshu et al., 2022; Delavar et al., 2019; Li et al., 2022). These encompass techniques such as transformation of input data, training and testing splits, feature selection, hyperparameter tuning, and cross-validation (refer to previous sections for details). These measures aim to constrain the uncertainties inherent in model development processes and fortify the model's predictive capabilities, for example by refining the interpretability of input data, mitigating the risk of overfitting, enhancing generalization performance, and minimizing the introduction of potentially noisy predictors.

In addition to the commonly adopted strategies in using XGBoost and the other ML techniques, we augment the control of model uncertainty through a representativeness check. This check ensures alignment between the distribution of model parameters used during training and those applied in predictions. This additional step serves to enhance the model's transferability from the training catchment to the broader CONUS domain. To gauge the representativeness of our chosen predictors, we conducted a Cumulative Distribution Function (CDF) comparison for each parameter between the observational dataset (derived from 2595 independent catchments) and the entire CONUS dataset (comprising approximately 2.6 million local catchments in NHDPlus). For this comparison, we assess the relative difference in the 5th, 25th, 50th, 75th, and 95th percentiles between the two CDFs. As an illustration, the relative difference for the 5th percentile is computed as the ratio of the difference between the 5th percentile of the available $P_r$ data and that of the entire CONUS data to their average. Table 2 provides a summary of the CDF comparison of the 15 selected predictors (Supplementary Fig. S6). A predictor is deemed representative of the whole CONUS if the average relative difference is less than 0.75. Following Abeshu et al. (2022), the choice of the 0.75 threshold strikes a balance between maintaining data representativeness and avoiding the exclusion of too many predictors. Three predictors, namely "BASIN_AREA", "NLCD01_95", and "NLCD01_52", have failed the representativeness check and are consequently excluded. Note that the ML model performance has only slightly changed after reducing the number of predictors from 15 to 12, as shown in Supplementary Fig. S7. Following the same process, the SoilGrids-based model excludes "NLCD01_95" during the representativeness check, resulting in 12 out of 13 predictors being retained for the final optimal model (Supplementary Table S5).

**Table 2. Representativeness of XGBoost model input predictors over CONUS.**

| Attributes | Relative difference in percentiles between $P_r$-*available* and *whole_conus* data | | | | | Average |
| | 5th | 25th | 50th | 75th | 95th | |
|---|---|---|---|---|---|---|
| BASIN_AREA | 1.941 | 1.728 | 1.669 | 1.794 | 1.900 | 1.806 |
| NLCD01_95 | 0.667 | 0.667 | 0.842 | 1.144 | 1.529 | 0.969 |
| NLCD01_52 | 0.353 | 0.624 | 1.224 | 1.482 | 0.889 | 0.914 |
| CNPY11_BUFF100 | 1.684 | 1.090 | 0.427 | 0.080 | 0.078 | 0.672 |
| NLCD01_90 | 0.769 | 0.314 | 0.461 | 0.621 | 0.807 | 0.594 |
| NLCD01_42 | 0.667 | 0.559 | 0.651 | 0.502 | 0.225 | 0.521 |
| elev_related | 0.769 | 0.806 | 0.320 | 0.621 | 0.008 | 0.505 |
| hydro_related | 0.584 | 0.898 | 0.316 | 0.108 | 0.106 | 0.402 |
| HGBD | 0.955 | 0.264 | 0.152 | 0.095 | 0.255 | 0.344 |
| CONTACT | 0.166 | 0.135 | 0.248 | 0.292 | 0.393 | 0.247 |
| BFI | 0.476 | 0.304 | 0.152 | 0.002 | 0.027 | 0.192 |
| RH | 0.197 | 0.103 | 0.015 | 0.014 | 0.014 | 0.068 |
| soil_texture_related | 0.095 | 0.071 | 0.068 | 0.071 | 0.015 | 0.064 |
| CWD | 0.063 | 0.065 | 0.028 | 0.053 | 0.033 | 0.048 |
| temp_related | 0.035 | 0.034 | 0.009 | 0.029 | 0.006 | 0.023 |

Abbreviations: BASIN_AREA (catchment area); NLCD01_95 (areal percentage of herbaceous wetlands); NLCD01_52 (areal percentage of shrub); CNPY11_BUFF100 (areal percentage of canopy in the riparian buffer); NLCD01_90 (areal percentage of woody wetlands); NLCD01_42 (areal percentage of evergreen forest); elev_related (merged predictor for mean/min/max elevation); hydro_related (merged predictor representing recharge, runoff, and precipitation); HGBD (areal percentage of Hydrologic Group BD soil); CONTACT (subsurface contact time); BFI (base flow index); RH (relative humidity); soil_texture_related (merged predictor for silt and sand content); CWD

(consecutive wet days); temp_related (merged predictor encompassing potential evapotranspiration, first/last freeze timing, snow fraction, actual evapotranspiration, and mean/min/max temperature); For detailed descriptions, refer to Supplementary Tables S2 and S3.

## 4.2 Sensitivity analyses

Model sensitivity analysis involves probing the importance of uncertainties in model parameters (Loucks and Van Beek, 2017). We examine our model's sensitivity to each selected predictor using two different methods: 1) dropping one predictor at a time and tracking the changes in model performance, and 2) the Sobol sensitivity analysis approach (Sobol, 2001). Figure 8 demonstrates the model performance difference in the training and testing phases after dropping one of the 12 variables. A 5% threshold is chosen to determine the significance of the change. In general, the shifting pattern in MASE scores remains consistent between the training and testing phases. However, the alterations in MASE values for most predictors, particularly during the testing phase, are minimal or even negligible. In other words, the model appears to be insensitive to most predictors according to this first sensitivity analysis method.

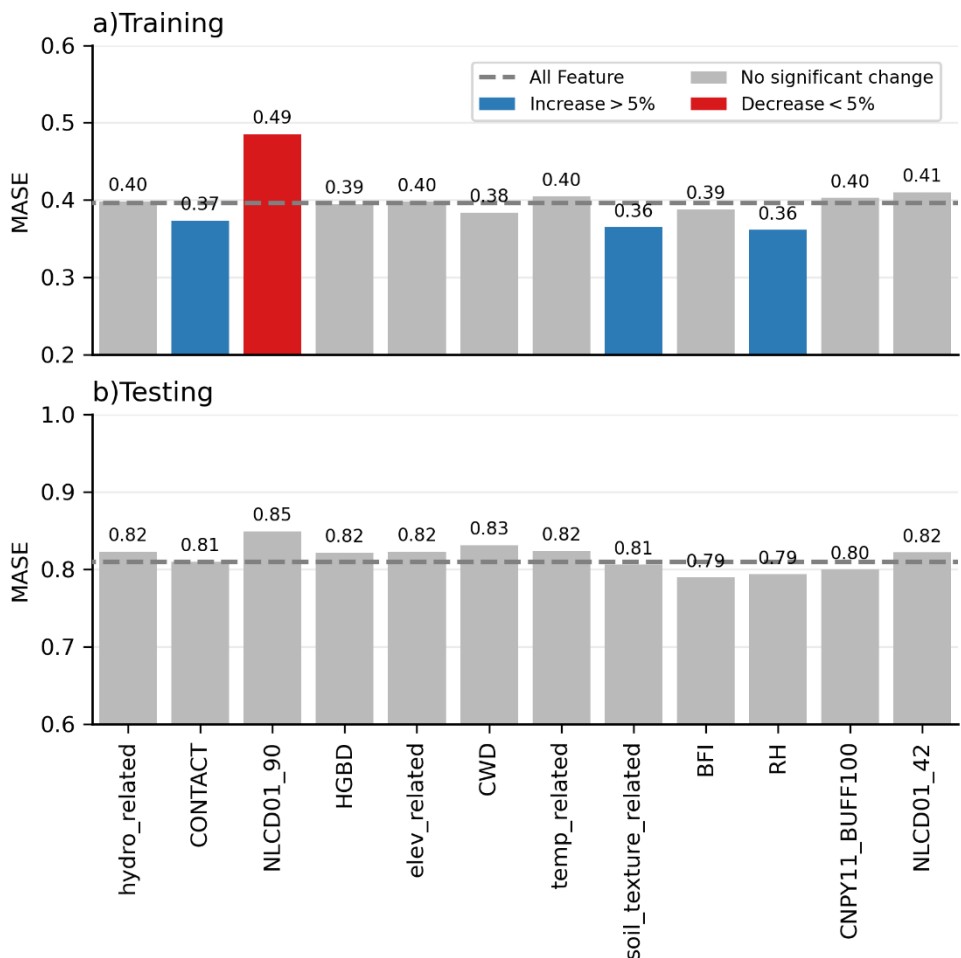

**Figure 8. Sensitivity of XGBoost model to predictors in the training and testing phases. The MASE value is represented by the blue, red, and grey bars, indicating whether the model performance increases, decreases, or remains relatively unchanged after dropping the corresponding predictor. The dashed grey line indicates the model performance with all variables included. Abbreviations: hydro_related (merged predictor representing recharge, runoff, and precipitation); CONTACT (subsurface contact time); NLCD01_90 (areal percentage of woody wetlands); HGBD (areal percentage of Hydrologic Group BD soil); elev_related (merged predictor for mean/min/max elevation); CWD (consecutive wet days); temp_related (merged predictor encompassing potential evapotranspiration, first/last freeze timing, snow fraction, actual evapotranspiration, and mean/min/max temperature); soil_texture_related (merged predictor for silt and sand content); BFI (base flow index); RH (relative humidity); CNPY11_BUFF100 (areal percentage of canopy in the riparian buffer); NLCD01_42 (areal percentage of evergreen forest). For detailed descriptions, refer to Supplementary Tables S2 and S3.**

The Sobol sensitivity analysis is a widely used variance-based global sensitivity analysis method (Borgonovo and Plischke, 2016). It provides two indices: First-order Index (S1), which measures the sensitivity of an individual predictor itself (local variance), and Total Index (ST), which accounts for the effects of both an individual predictor itself and its interactions with any other predictors (global variance) (Saltelli, 2002; Saltelli et al., 2010). These interactions, which can be of any order, can be isolated. For instance, second and higher-order interactions can be isolated by subtracting SI from ST. The results from the

Sobol test are summarized in Table 3. The distribution of S1 is highly right-skewed, suggesting that the model exhibits insensitivity to most predictors if only local variance is considered. There are, however, a few exceptions, such as "hydro_related", and "temp_related", which present high S1 values. The global variance, represented by the ST index, paints a somewhat different picture. When considering the ST index, a broad set of predictors emerge as sensitive, particularly those with ST values exceeding 0.1. It's worth noting that these predictors also hold high rankings in the predictor selection, as shown in Fig. 3. Furthermore, it is significant that 11 out of the total 12 predictors show a normalized difference between S1 and ST (calculated as (ST-S1)/ST) greater than 50%. This observation underscores the significant interactions among the predictors (Saltelli et al., 2010). This suggests that if a predictor is dropped, the remaining predictors could potentially compensate for its absence, highlighting the nonlinear, high-order interdependence among the predictors in our model.

**Table 3. Sobol sensitivity analysis results for the 12 selected predictors.**

| Predictors | Total Indices (ST) | First Order Indices (S1) | Difference ((ST-S1)/ST) |
|---|---|---|---|
| hydro_related | 0.466 | 0.291 | 0.375 |
| temp_related | 0.311 | 0.141 | 0.546 |
| CWD | 0.207 | 0.044 | 0.788 |
| CONTACT | 0.143 | 0.003 | 0.977 |
| CNPY11_BUFF100 | 0.132 | 0.028 | 0.787 |
| NLCD01_90 | 0.125 | 0.049 | 0.608 |
| elev_related | 0.087 | 0.017 | 0.806 |
| BFI | 0.072 | 0.012 | 0.831 |
| RH | 0.062 | 0.010 | 0.836 |
| soil_texture_related | 0.034 | 0.000 | 1.000 |
| NLCD01_42 | 0.024 | 0.005 | 0.798 |
| HGBD | 0.013 | 0.002 | 0.873 |

Abbreviations: hydro_related (merged predictor representing recharge, runoff, and precipitation); temp_related (merged predictor encompassing potential evapotranspiration, first/last freeze timing, snow fraction, actual evapotranspiration, and mean/min/max temperature); CWD (consecutive wet days); CONTACT (subsurface contact time); CNPY11_BUFF100 (areal percentage of canopy in the riparian buffer); NLCD01_90 (areal percentage of woody wetlands); elev_related (merged predictor for mean/min/max elevation); BFI (base flow index); RH (relative humidity); soil_texture_related (merged predictor for silt and sand content); NLCD01_42 (areal percentage of evergreen forest); HGBD (areal percentage of Hydrologic Group BD soil); For detailed descriptions, refer to Supplementary Tables S2 and S3.

The above sensitivity analyses suggest that our model exhibits low sensitivity to most predictors when considering their individual (local) impact. However, the Sobol sensitivity analysis uncovers a heightened degree of sensitivity in the context of global effects, particularly given the significant interactions among the predictors. A similar sensitivity analysis was conducted for the SoilGrids-based model, yielding the same conclusions (Supplementary Fig. S14 and Supplementary Table S6).

**5 Potential use and limitations**

The $P_r$ map has several promising uses. For instance, one of the pivotal applications of the $P_r$ map is to estimate the lateral
leaching of DOC. Figure 9, as an illustration, shows a $C_{DOC\_runoff}$ map over CONUS depicting the long-term average
concentration of DOC in the leaching flux at over two million NHDPlus local catchments. This map is derived based on Eqn.
(4), leveraging the $P_r$ map in Fig. 6 and the top-layer SOC data from HWDS1.2. Due to missing data in the HWSD 1km SOC
map at about 0.6 million NHDPlus local catchments, we cannot calculate the $C_{DOC\_runoff}$ values over those catchments.

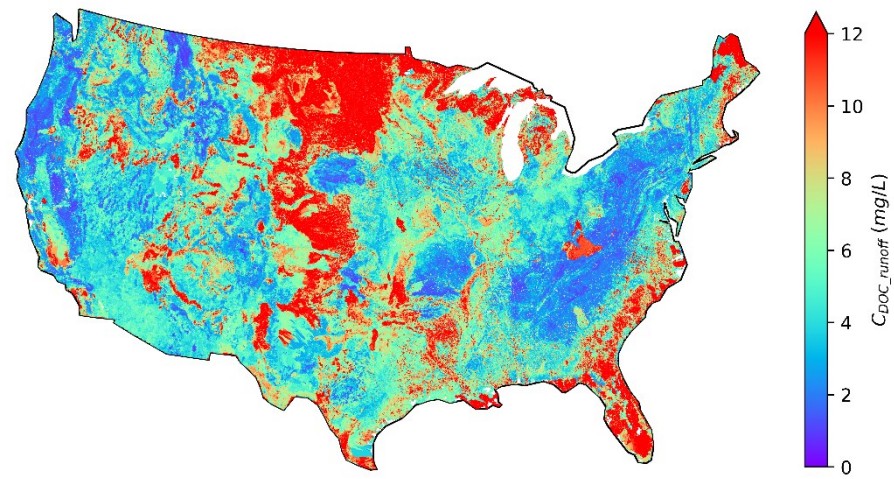

**Figure 9. Calculated CONUS map of DOC concentration in leaching flux from soils to over 2.6 million NHDPlus flowlines.**

The spatial patterns of the $C_{DOC\_runoff}$ map are highly correlated to those of the $P_r$ (Fig. 6) and SOC map (Supplementary Fig.
S5a). Notably, the $C_{DOC\_runoff}$ values are high in regions with extremely high SOC values. Additionally, the $C_{DOC\_runoff}$
values are high in North Dakota, Montana, and southern coasts, where the $P_r$ values are high. Interestingly, the influences of
$P_r$ and SOC can counterbalance each other in some places. For instance, in the upper Rocky Mountains, the SOC storage is
abundant due to the presence of forests. However, the low temperature in this region hinders microbial activities, resulting in
extremely low $P_r$ values. As a result, the concentration of DOC leaching flux is relatively low. Moreover, the spatial coverage
of wetlands also appears to be relevant (Supplementary Fig. S5b), which is consistent with the suggested crucial role of
wetlands in riverine DOC dynamics (Duan et al., 2017; Leibowitz et al., 2023). For instance, high $C_{DOC\_runoff}$ values are
observed in upper Minnesota, Florida, and Louisiana, where wetlands are prevalent. In places with few wetlands, like Nevada,
Arizona, and New Mexico, the leaching flux concentration is considerably lower.

There are at least two other potential uses of the $P_r$ map: 1) It can support large-scale DOC modeling over CONUS or a major river basin. For instance, testing the use of the map within the framework of the Energy Exascale Earth System Model (Burrows et al., 2020; Caldwell et al., 2019; Golaz et al., 2019) is ongoing and will be reported in the near future. 2) It can be used to provide a quick estimation of riverine DOC concentration or flux at any catchments where no DOC observations are available.

We caution the potential users of the $P_r$ map with several limitations in the methods invoked. Firstly, the $P_r$ values in the map account for the spatial heterogeneity of various DOC-related processes and factors only in a long-term average sense owing to the limited data availability, i.e., the SOC reanalysis data are long-term averages, and the observed riverine DOC data are only available at irregular time intervals. While we believe that such a $P_r$ map is a critical step in effectively capturing the spatial heterogeneity of the relevant processes and environmental factors, incorporating their temporal dynamics is beyond the scope of this study and left for future work. Second, the ML techniques are not process-based and thus do not yet offer rich insight into the relevant mechanisms. To improve our understanding of the DOC-related processes, the $P_r$ map should be used in conjunction with other observational data, process-based models, and carefully designed numerical experiments. Third, the lack of direct measurements of $P_r$ necessitates the use of indirect validation methods. To further enhance robustness, we encourage the design and implementation of new field experiments guided by our lumped parameter approach. Last but not least, the ML model has been trained with the data in the CONUS domain only, so it may not be transferable beyond CONUS.

Our lumped parameter approach and machine learning-based parameterization strategy are designed to generalize beyond the CONUS and scale globally. The framework is inherently generic, independent of site-specific characteristics, and supported by machine learning techniques adaptable to diverse regions. The CONUS study area, characterized by substantial spatial heterogeneity, provides a robust foundation for demonstrating this generalizability. However, extending the framework to a global scale introduces challenges, particularly in data availability and variability in environmental conditions. Addressing these requires extensive observational data collection, especially riverine DOC observations, leveraging public datasets, literature, and increased fieldwork for enhanced coverage. At the global scale, managing increased uncertainties is crucial, as larger variability is expected compared to the CONUS-based parameterization. Efforts should focus on assembling comprehensive catchment attributes while maintaining flexibility in their significance assessment, allowing the machine learning model to determine their importance contextually. High-priority attributes identified in this study (Fig. 3), such as woody wetland percentage, should receive particular attention as they are likely critical in other regions.

## 6 Data and code availability

The resulting $P_r$ and $C_{DOC\_runoff}$ maps over CONUS are freely available at https://zenodo.org/records/14563816 (Li et al., 2024). The Zenodo repository includes the following resources: a) Pr.gpkg – a 9.9 GB GeoPackage file containing data on Pr, SOC, and DOC, derived using SOC data from HWSD v1.2 and SoilGrids 2.0 across over 2.6 million NHDPlus local

catchments. This file also includes COMID and local catchment boundary polygons and is compatible with GIS software such as QGIS, ArcGIS, and Python libraries like GeoPandas for analysis and editing; b) PNG images – two high-resolution PNG files illustrating the HWSD-based and SoilGrids-based model-simulated Pr maps across over 2.6 million NHDPlus local catchments; c) Required input files – files necessary to reproduce the reported results; and d) ReadMe document – a text file providing detailed descriptions of each resource in the Zenodo repository. The input data are obtained from the water quality portal (https://www.waterqualitydata.us/), NHDPlus (https://www.epa.gov/waterdata/nhdplus-national-data), ScienceBase (https://doi.org/10.5066/F7765D7V), HWSD v1.2 (https://www.fao.org/soils-portal/data-hub/soil-maps-and-databases/harmonized-world-soil-database-v12/en/) and SoilGrids2.0 (https://files.isric.org/soilgrids/latest/data/). Additionally, the Python scripts used for feature selection, model training, and evaluation are available on the Github repository at https://github.com/Ceyxleo/DOC-Param-Map.

**7 Conclusions**

We developed two new maps of $P_r$, the transformation rate from SOC concentration in soil to DOC concentration in the leaching flux, over CONUS, based on SOC data from the HWSD v1.2 and SoilGrids 2.0. Evaluation of derived DOC concentrations at over 3000 WQP stations confirms the robustness of our methodology, which incorporates a generic formula linking SOC and DOC via $P_r$, riverine DOC observations, environmental variables, and ML techniques that effectively capture high-order, nonlinear relationships between $P_r$ and the environmental variables. These $P_r$ maps, the first of their kind, are highly valuable for large-scale DOC modeling and for improving our understanding of DOC-related processes across the land-river continuum.

**Author contributions**

LL performed the analysis with the inputs from the co-authors, prepared the figures, and wrote the first draft. HL devised the conceptual idea and supervised the study. GA provided frequent assistance in processing the data and developing the model. All the co-authors contributed to the writing.

**Competing interests**

At least one of the (co-)authors is a member of the editorial board of Earth System Science Data.

**Disclaimer**

Publisher's note: Copernicus Publications remains neutral with regard to jurisdictional claims in published maps and institutional affiliations.

**Acknowledgments**

This research is supported by the Office of Science of the U.S. Department of Energy Biological and Environmental Research
as part of the Earth System Model Development program area through the Energy Exascale Earth System Model (E3SM) project. The Pacific Northwest National Laboratory is operated by Battelle for the U.S. Department of Energy under Contract DE-AC05-76RL01830.

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
