# Peer review of "Transformation Rate Maps of Dissolved Organic Carbon in the Contiguous U.S."

_Earth System Science Data, 2024_

## Referee Comment (RC1)

**Review report for ESSD-2024-43**

'Deriving a Transformation Rate Map of Dissolved Organic Carbon over the Contiguous U.S.'

This study presents an innovative approach to deriving a high-resolution transformation rate (Pr) map of dissolved organic carbon (DOC) from soil organic carbon (SOC) across the contiguous United States using machine learning techniques, specifically XGBoost. By predicting DOC transformation rate based on various environmental attributes, the authors generate a DOC concentration reanalysis dataset for over two million small catchments. This research addresses the insufficient understanding of DOC conversion processes and provides a robust methodology for improving carbon cycle simulations in land surface and earth system models. The use of XGBoost to derive Pr values and create a high-resolution DOC concentration map is both novel and effective. The study's comprehensive data analysis and rigorous machine learning framework ensure robust and reliable results. Additionally, the findings have significant implications for enhancing carbon cycle models and informing climate change mitigation policies.

The article is logically structured, and the objectives are clear. The introduction is well-written and accessible, even to a geomorphologist like myself. According to the link provided in the 'Data availability' section, I downloaded all the Zenodo data. I opened and checked the raw data in ArcGIS, which includes SOC, PR, and DOC for the United States. The authors also provided a readme file explaining the attribute tables and their units.

If the authors can address my comments below and those of other reviewers through a major revision, the manuscript can be a valuable contribution to ESSD.

**Comments for manuscript**:
This database provides an excellent method for calculating DOC. However, it is limited to the United States. If other scholars wish to apply this method to other regions, the contribution of this paper would be even greater. Although the authors mention in the first paragraph of the Method section that 'The methodology here is described with specific details over the CONUS region, but it is transferable to other regions after some modifications based on data availability', they do not provide further details on how to apply this method to other regions. Especially since some environmental factors have been found to have significant impacts while others do not. This information is crucial for applications in other regions. It is suggested that the authors discuss this briefly in the Potential use section (how the methods and results of this paper can inspire the calculation of DOC or Pr in other regions globally).

The article is filled with numerous abbreviations, including those in the main text, figures, and tables, which increases the difficulty for readers. It is recommended that the authors avoid retaining so many abbreviations, especially those that appear infrequently in the later sections (e.g., fewer than five times) and those that are not used at all later on, for example, abbreviations like 'Pg' and 'ESMs' in the Introduction. In Tables 1, 2, 4, and 5, as a reader, I cannot understand what these parameters represent just by looking at the tables. Moreover, the parameter names in the 'Attributes' column of Table 1 are entirely unclear. Recommendations:

Avoid using abbreviations in the tables. If abbreviations must be used, provide explanations at the bottom of the table to help readers understand, rather than having them search the main text or supplementary information for the full terms. The content in the 'Attributes' column of Table 1 is completely incomprehensible. It is suggested to move Table 1 to the supplementary information.

The authors selected headwaters based on the following two criteria: 1) there are no upstream rivers flowing into them, and 2) their drainage areas are no more than 2500 km². I have the following questions:

1)Does the first criterion mean that the selected station can only have one river upstream without any tributaries?

2)If so, the second criterion excludes drainage areas larger than 2500 km². I find it hard to believe that a watershed of several thousand square kilometers has only one river without any tributaries. Please provide more details in the main text to clarify this and avoid confusion for readers like me.

3)Additionally, according to Fig. S1, the minimum drainage area in NHDPlus is 0.001 km², approximately a 30-meter square. As a geomorphologist, I do not understand how a watershed of this size can have sufficient upstream drainage area to form a river. Generally, river sources are not at the drainage divide but are 500-5000 meters downstream from it.

4)Moreover, I hope the authors clarify in the caption of Fig. S1 whether the data source includes all watersheds in NHDPlus or only the several thousand watersheds used in this study.

The authors need to explain in the main text the format and size of all the files uploaded to Zenodo, especially detailing what information is included in the files with the format 'gpkg' and what software readers can use to open and edit them.

Among the 29,320 WQP stations, some stations do not have existing upstream watershed boundaries. In such cases, the authors obtained the watershed boundaries using DEM. I have the following questions regarding this:

1) The authors need to clarify, among the 22,201 stations, how many watershed boundaries were derived from DEM and how many from NHDPlus?

2) What resolution of DEM was used, and how were the watershed boundaries calculated?

3) The authors should compare their calculated watershed boundaries with the global watershed boundaries based on a 90-m resolution DEM and advanced algorithms (ESSD 6, 1151–1166, 2024) and present a comparison figure in the SI.

The authors should provide a clear definition of "headwater" as used in this paper in the introduction. Is it determined based on stream order, river length, drainage area, or the number of tributaries? Additionally, they need to explain why they focus on headwaters.

Maybe convert Table 2 into a bar chart and place it in SI. Additionally, again, many abbreviations in Table 2 need to be explained with their full terms.

The language in this article needs further refinement. Here are just some examples that need to be revised, and the authors should check the entire text:

1) Delete "quickly" from line 149.

2) Delete "required for this study" from line 153.

3) Refine "We collect a wide range of environmental variables, comprising a total of 126 variables" to "We collect 126 environmental variables.".

4) Change "The ML technique used in this study is the eXtreme Gradient Boosting (XGBoost) algorithm" to "We use the eXtreme Gradient Boosting (XGBoost) ML algorithm."

The title is a bit long; it is recommended to change it to: "U.S. Transformation Rate Map of Dissolved Organic Carbon" or "Transformation Rate Map of Dissolved Organic Carbon in the Contiguous U.S."

The citation format for figures is completely inconsistent throughout the text. Examples for the same figure include: Fig. S1, supplementary Fig. S1, and Supplementary Fig. S1. Please check the entire text (main text, figures, SI) and standardize according to ESSD requirements.

L175 ScienceBase also provides indicators of human activities, right?

L244 "Out of the remaining 95 variables (see supplementary Tables S1 and S2 for details), 46 are relatively independent from each other. However, the other 49 are highly correlated with one or more variables." How did the authors determine "relatively independent" and "highly correlated"? I expect to see more explanation of this in the main text.

Line 249, change "see Supplementary Figure S3" to "Supplementary Figure S3." Please check the entire text for similar instances where "see" is unnecessary.

Line 251: "This new variable is thus independent of the other environmental variables." I do not understand the basis for this statement. Even if the new 9 combined parameters are formed, it is unlikely that they are completely independent of the other 46 parameters. The authors should provide a brief explanation in the main text or delete this sentence.

Lines 273-275 need to be supported by references.

Line 379: "per_canopy" is too difficult to understand.

In some places, it is written as "section," while in others, it is abbreviated as "sect" (e.g., L380).

L413 'Note the unit of DOC concentration in water is mostly reported in mg/L (Schelker et al., 2012; Tian et al., 2015b; Langeveld et al., 2020)'. I think this sentence is not important to be in the main text.

L481-482 'Blue, red, and grey colors are employed to indicate whether dropping the

corresponding predictor will result in an increase, decrease, or insignificant change in the model's performance, respectively' should be in figure caption, rather than here.

**Comments for dataset in Zenodo**:
There are many blank "nodata" areas within the CONUS_DOC_MAP, whereas the CONUS_PR_MAP does not have this issue. The authors need to explain this in the main text.

For reproducibility, the authors need to provide the shapefiles (or other similar vector data) for the 2595 watersheds used for machine learning training and the 3210 watersheds used for evaluation, as well as the shapefiles for these 5805 stations. The machine learning codes, as well as the raw data used for training the machine learning model, need to be uploaded to Zenodo; Then provide another link in the manuscript (not https://doi.org/10.5281/zenodo.8339372).

**Suggestion for figures:**
The background color of all 2D density plots needs to be changed because the background color is included in the color scale. This makes it difficult for readers to distinguish between the data and the background color.

Are the points in Figure 1 outlets or geometric centers of the watersheds? Additionally, it is necessary to indicate in the figure or caption that the gray lines represent rivers and the black lines represent national boundaries. Also, please specify the sources of these two elements.

Figures 4 and 7 contain numerous abbreviations that are not explained in the captions, making it difficult for readers to understand the figures directly.

It is necessary to explain in the caption of Figure 4 what the correlation coefficient is. Is it Spearman rank?

Why are there many nodata areas near the national boundaries in Figure 5?

Fig. S1 needs ticks on the X-axis.

In the main manuscript, I do not understand the differences between the two types of watershed boundaries provided by NHDPlus. Besides, I do not understand Figure S2. It is recommended to use a real terrain example for illustration to show the differences between these two kinds of watershed boundaries. For example, based on Google Earth, mark the river, the two different watersheds, and the DOC station location (outlet).

I don't have research experience with DOC; most of my comments are from a geomorphological perspective, as well as regarding readability and clarity. I hope my suggestions are helpful.

Chuanqi He      MIT, USA  21 May 2024

---

## Author Comment (AC2)

Reviewer #1:

This study presents an innovative approach to deriving a high-resolution transformation rate (Pr) map of dissolved organic carbon (DOC) from soil organic carbon (SOC) across the contiguous United States using machine learning techniques, specifically XGBoost. By predicting DOC transformation rate based on various environmental attributes, the authors generate a DOC concentration reanalysis dataset for over two million small catchments. This research addresses the insufficient understanding of DOC conversion processes and provides a robust methodology for improving carbon cycle simulations in land surface and earth system models. The use of XGBoost to derive Pr values and create a high-resolution DOC concentration map is both novel and effective. The study's comprehensive data analysis and rigorous machine learning framework ensure robust and reliable results. Additionally, the findings have significant implications for enhancing carbon cycle models and informing climate change mitigation policies.

The article is logically structured, and the objectives are clear. The introduction is well-written and accessible, even to a geomorphologist like myself. According to the link provided in the 'Data availability' section, I downloaded all the Zenodo data. I opened and checked the raw data in ArcGIS, which includes SOC, PR, and DOC for the United States. The authors also provided a readme file explaining the attribute tables and their units.

If the authors can address my comments below and those of other reviewers through a major revision, the manuscript can be a valuable contribution to ESSD.

**Response**: We greatly appreciate the thorough and insightful review of our manuscript. We will carefully address the specific comments, which we believe will substantially improve our manuscript. In the following, the reviewer's comments, our point-to-point responses, and our proposed revision text are shown in black, blue, and purple colors, respectively.

**Comments for manuscript:**

**Comment**: This database provides an excellent method for calculating DOC. However, it is limited to the United States. If other scholars wish to apply this method to other regions, the contribution of this paper would be even greater. Although the authors mention in the first paragraph of the Method section that 'The methodology here is described with specific details over the CONUS region, but it is transferable to other regions after some modifications based on data availability', they do not provide further details on how to apply this method to other regions. Especially since some environmental factors have been found to have significant impacts while others do not. This information is crucial for applications in other regions. It is suggested that the authors discuss this briefly in the Potential use section (how the methods and results of this paper can inspire the calculation of DOC or Pr in other regions globally).

**Response**: Thank you for your valuable feedback and for highlighting the importance of extending our methodology to regions beyond the United States. We will add the following discussion regarding applying our methodology to other regions in the "Potential Use" Section:
"This methodology framework, in principle, can be extended to regions beyond CONUS. However, the specific implementations may vary from one region to another, depending on the availability and quality of the input data, particularly DOC observations and associated catchment attributes. When gathering catchment attributes, it is advisable to collect as many attributes as possible. During the selection process, it is recommended to avoid making independent assumptions and instead allow the model to determine important attributes. More Attention could be given to the attributes listed in Table 2 of this study, especially those with high feature importance rankings, such as the Subsurface flow contact time and

woody wetland percentage. Nonetheless, it is important to recognize that the representativeness and relative importance of attributes may vary by location."

**Comment**: The article is filled with numerous abbreviations, including those in the main text, figures, and tables, which increases the difficulty for readers. It is recommended that the authors avoid retaining so many abbreviations, especially those that appear infrequently in the later sections (e.g., fewer than five times) and those that are not used at all later on, for example, abbreviations like 'Pg' and 'ESMs' in the Introduction. In Tables 1, 2, 4, and 5, as a reader, I cannot understand what these parameters represent just by looking at the tables. Moreover, the parameter names in the 'Attributes' column of Table 1 are entirely unclear. Recommendations: Avoid using abbreviations in the tables. If abbreviations must be used, provide explanations at the bottom of the table to help readers understand, rather than having them search the main text or supplementary information for the full terms. The content in the 'Attributes' column of Table 1 is completely incomprehensible. It is suggested to move Table 1 to the supplementary information.

**Response**: Great point on the use of abbreviations in our paper. We will follow your suggestions and make the following changes:
1. We will replace the abbreviations that are infrequently used throughout the paper with their full spellings.
2. While the abbreviated attribute names are directly adopted from NHDPlus (hence, it is not practical to completely avoid using them), we will add explanations of each abbreviation in the footnote of each table and in the caption of each figure to enhance the readability.
3. We will move Table 1 to Supplementary to better streamline the main text.

**Comment**: The authors selected headwaters based on the following two criteria: 1) there are no upstream rivers flowing into them, and 2) their drainage areas are no more than 2500 km². I have the following questions:
1)Does the first criterion mean that the selected station can only have one river upstream without any tributaries?
2)If so, the second criterion excludes drainage areas larger than 2500 km². I find it hard to believe that a watershed of several thousand square kilometers has only one river without any tributaries. Please provide more details in the main text to clarify this and avoid confusion for readers like me.
3)Additionally, according to Fig. S1, the minimum drainage area in NHDPlus is 0.001 km², approximately a 30-meter square. As a geomorphologist, I do not understand how a watershed of this size can have sufficient upstream drainage area to form a river. Generally, river sources are not at the drainage divide but are 500-5000 meters downstream from it.
4)Moreover, I hope the authors clarify in the caption of Fig. S1 whether the data source includes all watersheds in NHDPlus or only the several thousand watersheds used in this study.

**Response**: Great questions. It appears that the confusion has been caused by our loose definition of "headwater catchment". We did not use 'headwater catchment' following its strict geomorphological definition based on stream order, river length, drainage area, or the number of tributaries (He et al., 2024). Rather, we used "headwater catchment" for the drainage area extending from the WQP station upstream to the furthest tributaries that do not have any upstream rivers. To avoid this type of confusion, we will use "small catchment" instead of "headwater catchment" from now on. Below, we also provide specific answers to each of your questions:
   1)No, a small catchment can contain multiple tributaries.

2) It should be "small catchment". We will include the following definition of "small catchment" in Section 2.2 to prevent any confusion:
   "In this study, a "small catchment" refers to the drainage basin extending from the WQP station upstream to the furthest tributaries that do not have any upstream rivers. Note that a small catchment is not necessarily a headwater catchment which includes only one river (He et al., 2024)."
3) Most of the NHDPlus local catchments are not headwater catchments but intermediate catchments, serving as connections between other catchments. For example, they may correspond to only a very small river segment (as part of a longer river), making their size quite small.
4) We will change the caption of Fig. S1 to the following:
   "Figure S1. Distribution of the 2.6 million NHDPlus local catchment areas and flowline lengths."

**Comment**: The authors need to explain in the main text the format and size of all the files uploaded to Zenodo, especially detailing what information is included in the files with the format 'gpkg' and what software readers can use to open and edit them.

**Response**: We will add the following detailed information about all the files uploaded to Zenodo in the "Data Availability" Section:
"The Zenodo link contains four files: (1) **CONUS_PR_MAP.png**, a PNG image showing the model-simulated long-term averaged DOC transformation rate (Pr) across over 2.6 million NHDPlus local catchments; (2) **CONUS_DOC_MAP.png**, a PNG image depicting the long-term averaged DOC concentration reanalysis in soil leaching flux across over 2.6 million NHDPlus local catchments; (3) **DOC.gpkg**, a 9.9 GB GeoPackage file containing data on Pr and DOC for 2.6 million NHDPlus local catchments, including COMID, SOC concentration, and local catchment boundary polygons. This file is compatible with QGIS, ArcGIS, or Python libraries such as GeoPandas for opening and editing; and (4) **readme.txt**, a text file providing detailed information about the three aforementioned files."

**Comment**: Among the 29,320 WQP stations, some stations do not have existing upstream watershed boundaries. In such cases, the authors obtained the watershed boundaries using DEM. I have the following questions regarding this:
1) The authors need to clarify, among the 22,201 stations, how many watershed boundaries were derived from DEM and how many from NHDPlus?
2) What resolution of DEM was used, and how were the watershed boundaries calculated?
3) The authors should compare their calculated watershed boundaries with the global watershed boundaries based on a 90-m resolution DEM and advanced algorithms (ESSD 6, 1151–1166, 2024) and present a comparison figure in the SI.

**Response**: Thanks for these clarification questions. Below are our answers:
1) All watershed boundaries of the 22,201 stations were obtained using the HyRiver Python package (Chegini et al., 2021). This package does not directly derive catchment boundaries itself but retrieves them through The Hydro Network-Linked Data Index (NLDI) web server. Additionally, the package simplifies catchment boundaries and splits the catchment at the location of the WQP stations.
2) We did not use DEM to obtain watershed boundaries. We apologize for the incorrect information previously provided.
3) Since we did not derive the boundaries ourselves and the WQP stations can be located at any position along rivers (not necessarily at the outlet), it is not comparable with the global watershed boundaries based on a 90-m resolution DEM and advanced algorithms (ESSD 6, 1151–1166, 2024).
We will update the related paragraph as the following:

"We conduct a geospatial analysis to identify the upstream drainage area of each WQP river station using NHDPlus local catchments and flowlines. Utilizing the Python package HyRiver (Chegini et al., 2021), we co-located 29,320 WQP stations with the closest corresponding NHDPlus flowlines. However, 2,751 stations can not be linked due to the absence of adjacent flowlines. When WQP stations are in close proximity and share the same NHDPlus flowline, we retain only the station with the best data availability. For a given flowline, HyRiver traces back to every upstream flowline, accessing and merging the boundaries of all related NHDPlus local catchments (each flowline has its corresponding local catchment area) from the Hydro Network-Linked Data Index web server. It also requests the server to simplify boundaries and splits them precisely at the station locations. The relationship between the derived small catchment boundaries and the NHDPlus local catchments is shown in Figure S2a. Through this comprehensive geospatial analysis, we identify the upstream boundaries for 22,201 WQP stations."

**Comment**: The authors should provide a clear definition of "headwater" as used in this paper in the introduction. Is it determined based on stream order, river length, drainage area, or the number of tributaries? Additionally, they need to explain why they focus on headwaters.

**Response**: This comment has been essentially addressed in a previous response. We will use the terminology "small catchment" hereinafter, which refers to: a) DOC concentration at the outlet of a catchment is attributed to the entire upstream drainage area, and b) for small catchments, we can neglect the reaction of DOC in the stream at daily or sub-daily time steps, as explained in Section 2.1.

**Comment**: Maybe convert Table 2 into a bar chart and place it in SI. Additionally, again, many abbreviations in Table 2 need to be explained with their full terms.

**Response**: We will convert Table 2 into a bar chart, which we believe should remain in the main text as it highlights the importance of the selected features and is a critical result for Section 3.1. Additionally, the abbreviation issue has been addressed in a previous response by adding explanations to the figure captions and table footnotes.

**Comment**: The language in this article needs further refinement. Here are just some examples that need to be revised, and the authors should check the entire text:
1) Delete "quickly" from line 149.
2) Delete "required for this study" from line 153.
3) Refine "We collect a wide range of environmental variables, comprising a total of 126 variables" to "We collect 126 environmental variables.".
4) Change "The ML technique used in this study is the eXtreme Gradient Boosting (XGBoost) algorithm" to "We use the eXtreme Gradient Boosting (XGBoost) ML algorithm."

**Response**: We will carefully review and further refine the entire text. Regarding reviewer's specific comments:
1) We will delete "quickly" from line 149
2) We will delete "required for this study" from line 153
3) We will refine the original sentence to "We collect 126 environmental variables"
4) We will change the original sentence to "We use the eXtreme Gradient Boosting (XGBoost) ML algorithm"

**Comment**: The title is a bit long; it is recommended to change it to: "U.S. Transformation Rate Map of Dissolved Organic Carbon" or "Transformation Rate Map of Dissolved Organic Carbon in the Contiguous U.S."

**Response**: Great suggestion. We will change the title to "Transformation Rate Map of Dissolved Organic Carbon in the Contiguous U.S." to enhance clarity and conciseness.

**Comment**: The citation format for figures is completely inconsistent throughout the text. Examples for the same figure include: Fig. S1, supplementary Fig. S1, and Supplementary Fig. S1. Please check the entire text (main text, figures, SI) and standardize according to ESSD requirements.

**Response**: We will thoroughly review the entire manuscript, including the main text, figures, and supplementary information, to ensure that all figure citations are standardized according to the ESSD requirements. Specifically, we will consistently use the format "Fig. S1" throughout the manuscript.

**Comment**: L175 ScienceBase also provides indicators of human activities, right?

**Response**: Yes, you are correct. ScienceBase provides catchment attributes across 11 categories, including human activities. In our study, we present the four categories that we found significant in predicting DOC. We will refine the sentence as follows to avoid confusion:
"ScienceBase includes a wide range of environmental variables across 11 categories, such as climate, hydrology, soil, and geological data, conveniently available at the catchment scale across the entire CONUS."

**Comment**: L244 "Out of the remaining 95 variables (see supplementary Tables S1 and S2 for details), 46 are relatively independent from each other. However, the other 49 are highly correlated with one or more variables." How did the authors determine "relatively independent" and "highly correlated"? I expect to see more explanation of this in the main text.

**Response**: Thanks for pointing out this, we will add the following clarification:
"A "correlated group" is defined as a set of variables that are highly correlated, with a Pearson correlation coefficient $\geq 0.8$ or $\leq -0.8$. Within this group, every variable has at least one other variable that it is highly correlated with, and not with those outside the group. A variable is considered relatively independent if its correlation coefficient with all other variables is $< 0.8$ or $> -0.8$. The correlation threshold of $\pm 0.8$ is adopted following the guidelines by Schober et al. (2018)."

**Comment**: Line 249, change "see Supplementary Figure S3" to "Supplementary Figure S3." Please check the entire text for similar instances where "see" is unnecessary.

**Response**: We will change "see Supplementary Figure S3" to "Supplementary Figure S3" on line 249. Additionally, we will carefully review the entire text to identify and remove any unnecessary instances of "see" in figure citations to enhance clarity and consistency.

**Comment**: Line 251: "This new variable is thus independent of the other environmental variables." I do not understand the basis for this statement. Even if the new 9 combined parameters are formed, it is unlikely that they are completely independent of the other 46 parameters. The authors should provide a brief explanation in the main text or delete this sentence.

**Response**: Our original intent was to convey that after this treatment, the newly merged, single variable is considered to be relatively independent of the other environmental variables. We will revise the sentence as follows:
"This new variable is now relatively independent of the other environmental variables."

**Comment**: Lines 273-275 need to be supported by references.

**Response**: Those two sentences, "Recent studies have demonstrated the efficiency and effectiveness of these techniques in capturing high-dimensional and complex relationships between a target biogeochemical variable and various environmental predictors," and "These techniques have been successfully applied in various studies, including riverine sediment, beach water quality, oceanic particulate organic carbon, and eutrophication impacts from corn production (Abeshu et al., 2022; Li et al., 2022; Liu et al., 2021; Romeiko et al., 2020; Fan et al., 2021)," share the same citations. We will modify these two sentences as follows:
"These techniques have been successfully applied in various studies, including riverine sediment, beach water quality, oceanic particulate organic carbon, and eutrophication impacts from corn production (Abeshu et al., 2022; Li et al., 2022; Liu et al., 2021; Romeiko et al., 2020; Fan et al., 2021), demonstrating their efficiency and effectiveness in capturing high-dimensional and complex relationships between a target biogeochemical variable and various environmental predictors."

**Comment**: Line 379: "per_canopy" is too difficult to understand.

**Response**: This has been addressed in a previous comment. All the abbreviations in the "Name used in this study" column of Table 2 will be deleted. Instead, we will use the original names as provided in the NHDPlus dataset.

**Comment**: In some places, it is written as "section," while in others, it is abbreviated as "sect" (e.g., L380).

**Response**: Thanks. We will review the entire manuscript to ensure consistent usage of the term "section."

**Comment**: L413' Note the unit of DOC concentration in water is mostly reported in mg/L (Schelker et al., 2012; Tian et al., 2015b; Langeveld et al., 2020)'. I think this sentence is not important to be in the main text.

**Response**: Agree, it will be deleted from the main text.

**Comment**: L481-482 "Blue, red, and grey colors are employed to indicate whether dropping the corresponding predictor will result in an increase, decrease, or insignificant change in the model's performance, respectively" should be in figure caption, rather than here.

**Response**: We will move the explanation about the colors indicating changes in model performance from lines 481-482 to the caption in Figure 7.

**Comments for dataset in Zenodo:**

**Comment**: There are many blank "nodata" areas within the CONUS_DOC_MAP, whereas the CONUS_PR_MAP does not have this issue. The authors need to explain this in the main text.

**Response**: It has already been explained in the first paragraph of section 5 as: "Due to missing data in the HWSD 1km SOC map at about 0.6 million NHDPlus local catchments, we cannot calculate the $C_{DOC\_runoff}$ values over those catchments."

**Comment**: For reproducibility, the authors need to provide the shapefiles (or other similar vector data) for the 2595 watersheds used for machine learning training and the 3210 watersheds used for evaluation, as well as the shapefiles for these 5805 stations. The machine learning codes, as well as the raw data used for training the machine learning model, need to be uploaded to Zenodo; Then provide another link in the manuscript (not https://doi.org/10.5281/zenodo.8339372).

**Response**: Thank you. We will upload the shapefiles for the 2,595 watersheds used for machine learning training, the 3,210 watersheds used for evaluation, the 5,805 stations, the machine learning codes, and the raw data used for training the model to Zenodo. We will provide the new Zenodo link in the manuscript to ensure reproducibility.

**Suggestion for figures:**

**Comment**: The background color of all 2D density plots needs to be changed because the background color is included in the color scale. This makes it difficult for readers to distinguish between the data and the background color.

**Response**: Good catch! We will change the background color of all 2D density plots to light grey to ensure it is not included in the color scale.

**Comment**: Are the points in Figure 1 outlets or geometric centers of the watersheds? Additionally, it is necessary to indicate in the figure or caption that the gray lines represent rivers and the black lines represent national boundaries. Also, please specify the sources of these two elements.

**Response**: Thank you for the question. The points represent the locations of the WQP stations, which are also the outlets of the corresponding small catchments. The CONUS boundary was obtained from the GeoPandas built-in shapefile data, accessible through geopandas.read_file(gpd.datasets.get_path('naturalearth_lowres')).
The river shapefile was obtained from Natural Earth. We believe it is not very important to mention the source in the main text, as both are open-source data. We will revise the caption as follows:
"The points indicate the locations of the WQP stations, which are also the outlets of the corresponding small catchments. The CONUS boundary and river shapefiles are directly from open-source datasets GeoPandas (Van den Bossche et al., 2024 and Natural Earth (Made with Natural Earth. Free vector and raster map data @ naturalearthdata.com.), respectively."

**Comment**: Figures 4 and 7 contain numerous abbreviations that are not explained in the captions, making it difficult for readers to understand the figures directly.

**Response**: This issue has been addressed previously by adding explanations into the captions.

**Comment**: It is necessary to explain in the caption of Figure 4 what the correlation coefficient is. Is it Spearman rank?

**Response**: We will revise the caption of Figure 4 to specify that the Pearson correlation coefficient is used. The updated caption now reads:

"Figure 4. Covariance heatmap of Pr and the 12 selected NHDPlus predictors. The Pearson correlation coefficient is used."

**Comment**: Why are there many nodata areas near the national boundaries in Figure 5?

**Response**: Nice catch! This discrepancy is not due to missing data but rather a mismatch between two geo-datasets. The country boundaries, obtained from gpd.datasets.get_path("naturalearth_lowres") in the GeoPandas library, are of very low resolution. In contrast, the NHDPlus local catchments are derived from a 30m DEM, which are much more accurate. This difference in resolution leads to discrepancies at the national boundaries. Additionally, for approximately 14,000 NHDPlus local catchments, we cannot retrieve their catchment attributes, so they are removed from the prediction set.

**Comment**: Fig. S1 needs ticks on the X-axis.

**Response**: We will add ticks on the X-axis of Fig. S1.

**Comment**: In the main manuscript, I do not understand the differences between the two types of watershed boundaries provided by NHDPlus. Besides, I do not understand Figure S2. It is recommended to use a real terrain example for illustration to show the differences between these two kinds of watershed boundaries. For example, based on Google Earth, mark the river, the two different watersheds, and the DOC station location (outlet).

**Response**: Upon rechecking the NHDPlus version 2.1 National Seamless Geodatabase (.gdb), we found that it contains only the boundaries of local catchments. We apologize for the incorrect information provided earlier, and will update the corresponding paragraphs in the main text. To enhance clarity, we will regenerate Figure S2 to include two subplots: a) demonstrating the relationship between small catchments and the NHDPlus local catchments within them, and b) illustrating the nesting of small catchments and how we handle them.

**Comment**: I don't have research experience with DOC; most of my comments are from a geomorphological perspective, as well as regarding readability and clarity. I hope my suggestions are helpful.

**Response**: Yes, your suggestions are indeed very helpful for improving the readability and clarity. We are very grateful.

---

## Author Comment (AC5)

Reviewer #3:

The manuscript addresses a significant issue in the field of environmental science, particularly in understanding the dynamics of dissolved organic carbon (DOC) in the context of climate change and carbon cycling. The manuscript makes effective use of the public available datasets, especially the Water Quality Portal (WQP), provide a solid foundation for the analysis. The proposed method of estimating transformation rates from soil organic carbon (SOC) to DOC using a lumped parameter approach is innovative and could simplify large-scale modeling efforts. The model's simplicity and the reduced data requirements are strengths, making it more accessible for application in regions with limited data availability. And lastly, the model's potential to predict riverine DOC concentrations from SOC values is a valuable tool for water quality management and environmental monitoring. However, there are some potential weaknesses for the authors to consider and to improve the quality of the manuscript: (1) Generalizability: The study focuses on the contiguous U.S., and it is unclear how well the findings and models could be generalized to other regions with different environmental conditions. (2) Complexity of DOC Dynamics: The simplification of the model might overlook the complexity of DOC dynamics, including the influence of various biotic and abiotic factors. (3) Validation and Calibration: The manuscript would benefit from a more detailed discussion on the validation and calibration of the model, including the use of independent datasets. (4) Potential Over-simplification: The assumption that riverine DOC degradation in headwater streams is negligible might be an oversimplification, especially in ecosystems with high microbial activity. (5) Lack of Experimental Data: The study relies heavily on existing datasets, and there is a lack of experimental data to support the model's predictions. Overall, the development of a predictive model that can estimate riverine DOC concentrations from SOC values is innovative and has practical applications, I would recommend the manuscript for acceptance with major revision.

**Response**: We appreciate the reviewer's insightful comments. In fact, many of these points were central during the planning and implementation phases of this study. It is common for modelers to face a dilemma between complexity and simplicity. According to the principle of Occam's Razor (Walsh, 1979), complexity does not always bring better model predictive power, particularly in cases where the scientific community doesn't yet have a clear understanding of the relevant processes. In our case, it is our observation that the land modeling and biogeochemical science communities have not yet achieved a clear understanding of the DOC dynamics, as evidenced by the inconsistent descriptions of DOC leaching processes in existing models, such as DLEM, INCA-C, JULESDOCM, ECO3D, and TRIPLEX-HYD. More specifically, the communities are still unclear about 1) how many specific processes are involved from the land to aquatic ecosystems regarding DOC dynamics, 2) whether our understanding of each specific process is clear enough to allow robust mathematical expressions (aka, governing equations), and 3) how we can parameterize each governing equation to effectively account for spatiotemporal heterogeneities in the relevant controlling factors. It is also our observation that the currently available observations (known variables) are too limited comparing to the number of parameters in existing models (unknown variables) to enable parsimonious process descriptions, i.e., overparameterization. For instance, for modeling riverine DOC at the regional and larger scales, to the best of our knowledge, the only observation data available at the corresponding scales are the DOC observations from the river gauges. Based on these rationales,

we proposed our simplified formula with the hope that it is complementary to the existing, pioneering modeling approaches. As the reviewer rightfully pointed out, this "lumped parameter approach" has the advantages of "simplicity and the reduced data requirements", allowing for the usage of machine-learning techniques in the parameterization strategy. More importantly, the resulting parameter map is indeed effective, as demonstrated in our other ongoing modeling study, where we used the parameter map as a key input to a land-river modeling framework for DOC, validated the model simulated riverine DOC concentration values against the observed at over 450 large river gauges over the contiguous U.S., where the drainage areas of the gauges range from 55 km$^2$ to 1.1x10$^6$ km$^2$. Our modeling results are still preliminary since we are still adding and debugging the coding of other relevant processes, but both R-square and Kling-Gupta efficiency exceed 0.6 already, suggesting the fidelity of the parameterization strategy in the context of regional-scale DOC modeling. That said, given that our current study is mainly a dataset development effort tailored for ESSD, not a full-scale modeling one, we will report our modeling study in a separate manuscript.

[Figure]

**Figure R1**: Spatial distribution of DOC over 450 river gauges with DOC observations (top) and comparison between the simulated and observed long-term average riverine DOC concentrations at these stations (bottom). *Note: We provide these figures here only as part of our responses to*

*the reviewer's comments. It is NOT our intention to publish these figures as part of the manuscript under review here.*

Next we provide point-to-point responses to each major comment:

1) We believe that our lumped parameter approach and machine learning-based parameterization strategy are generalizable to other regions. The conceptualization of the lumped parameter approach itself is generic and not site-specific. The machine learning techniques are not site-specific either. Moreover, the contiguous U.S. as a study area itself contains significant spatial heterogeneities of environmental conditions, including diverse vegetation types, soil compositions, topographic variations, and climate regimes, that can be found elsewhere. In fact, one of our next steps is to expand our methodology framework over the global domain and produce a global parameter map, which will be reported in a separate study. At the global scale, the data availability is understandably less than the U.S. Our tentative plan to overcome this limitation include but not limited to: 1) collect as much as possible observational data, particularly riverine DOC observations, from public datasets and literature taking advantage of modern AI techniques; 2) call for more field work to collect DOC observations; 3) caution the unavoidably larger uncertainties embedded in the global parameter map (comparing to the U.S. map). We will add some discussions about the generalizability of our ML-based parameterization in the revised manuscript.

2) We are well aware of the complexity of DOC dynamics, particularly regarding lateral leaching processes in soil. These processes are influenced by numerous biotic factors, including microbial decomposition, plant root exudation, and enzymatic activities, as well as abiotic factors such as soil temperature, moisture, pH, and sorption-desorption processes. Even current process-based DOC models do not capture all these mechanisms but rather implement various, simplified descriptions of them, which in turn require extensive parameterizations as we discussed previously. Therefore, we propose our study as a first step towards a new pathway to advance the understanding of DOC dynamics that is complementary to the existing modeling approaches. We will further emphasize this point in the revised manuscript.

3) We agree that it is important to have independent datasets for calibration and validation. It appears that we may not have provided a clear description about the validation dataset. Our validation catchments are NOT within but encompass the catchments we used for the ML modeling. Therefore, the validation strategy we applied is appropriate for a dataset study. We will revise the methodology section to describe more clearly which datasets were used for calibration and which datasets were used for validation. We will further expand the discussion on the validation and calibration for better clarity.

4) We respectfully argue that our assumption is valid for most, if not all, headwater streams. There are two rationales behind our assumption: 1) Based on a literature review, we summarized the DOC degradation rates used in existing process-based modeling studies and reported by the experimental studies, as shown in Table R1 below. All of these studies suggest that, for headwater streams, the in-stream DOC degradation rate is approximately 0.01 per day; 2) Typical residence time of DOC in headwater streams (from the moment it enters into streams from soils to the moment it leaves the headwater streams into downstream rivers) is on the order of a couple of hours, i.e., much less than a

day (Ducharne et al., 2003; Li et al., 2013). Taken together, it is reasonable to assume that the DOC degradation is negligible between the moment it enters the streams from soils and the moment it leaves the headwater streams, hence supporting Eqn. (5). That said, we will carry out more literature review during the revision stage. Suppose we can find some studies documenting high instream DOC degradation rates in some headwater streams (i.e., with high microbial activity). In that case, we will evaluate the representativeness of such headwater streams globally and design a corresponding strategy for accounting for them in our study. At the very least, we will add some further discussion on this assumption and potential directions to improve it in the future.

Table R1. In-stream DOC degradation Rate ($k$) from previous modeling and experimental studies

| Study Type | First-Order Decay rate ($k$ d$^{-1}$) | Study Domain | Reference |
|---|---|---|---|
| Modeling | 0.01 | Eastern North America | Tian et al., 2015 |
| | 0.01 | Global | Li et al., 2019 |
| | 0.0163/0.0223[a] | Upland and forested catchments in Canada | Futter et al., 2007 |
| Experimental | 0.011[b] | Upland and forested catchment (Southern Appalachian Mountains, USA) | Qualls and Haines, 1992 |
| | 0.009[b] | Upland and forested catchment (Catskill Mountains, USA) | Sobczak et al., 2003 |
| | 0.013[c] | Forested headwater catchment (Haean Basin, South Korea) | Jung et al., 2014 |
| | 0.09[c] | Agro-urban headwater catchments (Taihu Lake Watershed, China) | Wu et al., 2019 |

*a. calibrated for the two catchments separately.*
*b. adopted from Table 2 in Mineau et al. (2016).*
*c. calculated by fitting a first-order decay model using the published data.*

5) On the one hand, we believe that observed riverine DOC data are already a very reliable source of validation data. On the other hand, we suggest that new field experiments could be designed and implemented following our lumped parameter approach, which innovatively provides a direct linkage between the land and river carbon pools. We will add a few sentences in the discussion addressing this concern.

**In addition, I have a few minor comments; please see below:**

**Response**: These minor comments are quite helpful as well. We provide our point-to-point responses to them, in blue color, in the following.

**Comment**: Line 131: "Eqn. (4) has several advantages" change to "Eqn. (4) has two advantages".

**Response**: We will change to "Eqn. (4) has two advantages".

**Comment**: Line 153: There are much higher spatial resolution SOC data available (e.g. SoilGrids provides 250m resolution data available, see reference below), why chose use HWSD?

Hengl, Tomislav, Jorge Mendes de Jesus, Gerard BM Heuvelink, Maria Ruiperez Gonzalez, Milan Kilibarda, Aleksandar Blagotić, Wei Shangguan et al. "SoilGrids250m: Global gridded soil information based on machine learning." PLoS one 12, no. 2 (2017): e0169748.

**Response**: Thanks for pointing it out. HWSD is a well-established dataset that has been extensively used in earth system modeling. That said, in our revision, we will also carefully consider using the SoilGrids250m dataset and comparing the results with those using HWSD.

**Comment**: Line 219-220: According to the description, 3210 pairs for evaluation are within the catchment of 2595 pairs for ML modeling, therefore they are not independent and the evaluation might biased.

**Response**: The evaluation catchments are not within the independent catchments, but rather, they encompass the independent catchments. In cases of paired & nested catchments, we take the one with a smaller drainage area for developing our ML modeling, and leave the one with a larger drainage area for future validation. Hence, our validation strategy is effective. To avoid further confusion, we will update Figure S2 to reflect the actual boundaries of nested catchments and revise the text accordingly.

**Comment**: Figure 3: In scatter plots, observed data are typically placed on the y-axis, while simulated data are positioned on the x-axis. I suggest moving the estimated Pr to the y-axis and the simulated Pr to the x-axis. The same recommendation applies to Figure 6.

**Response**: We have looked into several recently published articles in ESSD, and found that, observed data are mostly placed on the horizontal x-axis instead. Therefore, we will respectfully keep our current axis arrangement.

**Reference:**

Ducharne, A., Golaz, C., Leblois, E., Laval, K., Polcher, J., Ledoux, E., & de Marsily, G. (2003). Development of a high resolution runoff routing model, calibration and application to assess runoff from the LMD GCM. *Journal of Hydrology*, *280*(1-4), 207-228.

Futter, M. N., Butterfield, D., Cosby, B. J., Dillon, P. J., Wade, A. J., & Whitehead, P. G. (2007). Modeling the mechanisms that control in-stream dissolved organic carbon dynamics in upland and forested catchments. *Water Resources Research*, *43*(2).

Jung, B. J., Lee, J. K., Kim, H., & Park, J. H. (2014). Export, biodegradation, and disinfection byproduct formation of dissolved and particulate organic carbon in a forested headwater stream during extreme rainfall events. *Biogeosciences*, *11*(21), 6119-6129.

Li, H., Wigmosta, M. S., Wu, H., Huang, M., Ke, Y., Coleman, A. M., & Leung, L. R. (2013). A physically based runoff routing model for land surface and earth system models. *Journal of Hydrometeorology*, *14*(3), 808-828.

Li, M., Peng, C., Zhou, X., Yang, Y., Guo, Y., Shi, G., & Zhu, Q. (2019). Modeling global riverine DOC flux dynamics from 1951 to 2015. *Journal of Advances in Modeling Earth Systems*, *11*(2), 514-530.

Mineau, M. M., Wollheim, W. M., Buffam, I., Findlay, S. E., Hall Jr, R. O., Hotchkiss, E. R., ... & Parr, T. B. (2016). Dissolved organic carbon uptake in streams: A review and assessment of reach-scale measurements. *Journal of Geophysical Research: Biogeosciences*, *121*(8), 2019-2029.

Qualls, R. G., & Haines, B. L. (1992). Biodegradability of dissolved organic matter in forest throughfall, soil solution, and stream water. *Soil Science Society of America Journal*, *56*(2), 578-586.

Sobczak, W. V., Findlay, S., & Dye, S. (2003). Relationships between DOC bioavailability and nitrate removal in an upland stream: an experimental approach. *Biogeochemistry*, *62*, 309-327.

Tian, H., Yang, Q., Najjar, R. G., Ren, W., Friedrichs, M. A., Hopkinson, C. S., & Pan, S. (2015). Anthropogenic and climatic influences on carbon fluxes from eastern North America to the Atlantic Ocean: A process-based modeling study. *Journal of Geophysical Research: Biogeosciences*, *120*(4), 757-772.

Walsh, D. (1979). Occam's razor: A principle of intellectual elegance. American Philosophical Quarterly, 16(3), 241-244.

Wu, Z., Wu, W., Lin, C., Zhou, S., & Xiong, J. (2019). Deciphering the origins, composition and microbial fate of dissolved organic matter in agro-urban headwater streams. *Science of the total environment*, *659*, 1484-1495.

---

## Author Response (AR1)

In the following, the reviewer's comments, our point-to-point responses, our revisions are shown in black, blue, and purple colors, respectively.

**Reviewer #1:**

This study presents an innovative approach to deriving a high-resolution transformation rate (Pr) map of dissolved organic carbon (DOC) from soil organic carbon (SOC) across the contiguous United States using machine learning techniques, specifically XGBoost. By predicting DOC transformation rate based on various environmental attributes, the authors generate a DOC concentration reanalysis dataset for over two million small catchments. This research addresses the insufficient understanding of DOC conversion processes and provides a robust methodology for improving carbon cycle simulations in land surface and earth system models. The use of XGBoost to derive Pr values and create a high-resolution DOC concentration map is both novel and effective. The study's comprehensive data analysis and rigorous machine learning framework ensure robust and reliable results. Additionally, the findings have significant implications for enhancing carbon cycle models and informing climate change mitigation policies.

The article is logically structured, and the objectives are clear. The introduction is well-written and accessible, even to a geomorphologist like myself. According to the link provided in the 'Data availability' section, I downloaded all the Zenodo data. I opened and checked the raw data in ArcGIS, which includes SOC, PR, and DOC for the United States. The authors also provided a readme file explaining the attribute tables and their units.

If the authors can address my comments below and those of other reviewers through a major revision, the manuscript can be a valuable contribution to ESSD.

**Response**: We greatly appreciate the thorough and insightful review of our manuscript. We have carefully addressed the specific comments, which we believe will substantially improve our manuscript.

**Comments for manuscript:**

**Comment**: This database provides an excellent method for calculating DOC. However, it is limited to the United States. If other scholars wish to apply this method to other regions, the contribution of this paper would be even greater. Although the authors mention in the first paragraph of the Method section that 'The methodology here is described with specific details over the CONUS region, but it is transferable to other regions after some modifications based on data availability', they do not provide further details on how to apply this method to other regions. Especially since some environmental factors have been found to have significant impacts while others do not. This information is crucial for applications in other regions. It is suggested that the authors discuss this briefly in the Potential use section (how the methods and results of this paper can inspire the calculation of DOC or Pr in other regions globally).

**Response**: Thank you for highlighting the importance of extending our methodology to regions beyond the CONUS. While the overall framework is readily transferable to other regions, the primary challenges to a global application lie in the limited availability of DOC observation data and catchment attributes outside the CONUS. Therefore, we believe it is reasonable to leave a global-scale study for future work, when suitable data become available. In response, we have added a discussion in the "Potential Use" section (Lines 626-636) regarding address these challenges.

**Comment**: The article is filled with numerous abbreviations, including those in the main text, figures, and tables, which increases the difficulty for readers. It is recommended that the authors avoid retaining so many abbreviations, especially those that appear infrequently in the later sections (e.g., fewer than five times) and those that are not used at all later on, for example, abbreviations like 'Pg' and 'ESMs' in the Introduction. In Tables 1, 2, 4, and 5, as a reader, I cannot understand what these parameters represent just by looking at the tables. Moreover, the parameter names in the 'Attributes' column of Table 1 are entirely unclear. Recommendations: Avoid using abbreviations in the tables. If abbreviations must be used, provide explanations at the bottom of the table to help readers understand, rather than having them search the main text or supplementary information for the full terms. The content in the 'Attributes' column of Table 1 is completely incomprehensible. It is suggested to move Table 1 to the supplementary information.

**Response**: Great point on the use of abbreviations in our paper. To address this, we have made the following changes:

- 1. Abbreviations that are used infrequently throughout the paper have been removed and replaced with their full terms for improved clarity.
- 2. While the attribute names are directly adopted from NHDPlus and are therefore challenging to modify, we have added detailed explanations for each abbreviation in the footnotes of the relevant tables (Table 2, 3, S5 and S6) and captions of the figures (Figure 3, 5, 8, S6, S9, S11 and S14).
- 3. Table 1 has been removed from the main text. Detailed descriptions of the 46 originally relatively independent predictors and the 9 newly merged predictors from the correlation groups can now be found in Tables S2 and S3, respectively.

**Comment**: The authors selected headwaters based on the following two criteria: 1) there are no upstream rivers flowing into them, and 2) their drainage areas are no more than 2500 km2. I have the following questions:

1) Does the first criterion mean that the selected station can only have one river upstream without any tributaries?

2) If so, the second criterion excludes drainage areas larger than 2500 km2. I find it hard to believe that a watershed of several thousand square kilometers has only one river without any tributaries. Please provide more details in the main text to clarify this and avoid confusion for readers like me.

3) Additionally, according to Fig. S1, the minimum drainage area in NHDPlus is 0.001 km2, approximately a 30-meter square. As a geomorphologist, I do not understand how a watershed of

this size can have sufficient upstream drainage area to form a river. Generally, river sources are not at the drainage divide but are 500-5000 meters downstream from it.

4) Moreover, I hope the authors clarify in the caption of Fig. S1 whether the data source includes all watersheds in NHDPlus or only the several thousand watersheds used in this study.

**Response**: Great questions. It appears that the confusions were caused by our loose definition of "headwater catchment". We did not use 'headwater catchment' following its strict geomorphological definition based on stream order, river length, drainage area, or the number of tributaries (He et al., 2024). Rather, we used the term "headwater catchment" for the drainage area extending from the WQP station upstream to the furthest tributaries that do not receive inflow from other upstream rivers. To avoid this type of confusion, "small catchment" has been used instead of "headwater catchment". Below, we also provide specific answers to each of your questions:

- 1) Answer for Q1: No, a small catchment can contain multiple tributaries.
- 2) Answer for Q2: We have included the definition of "small catchment" (Lines 142-144) in Section 2.1 to prevent any confusion.
- 3) Answer for Q3: Most of the NHDPlus local catchments are not headwater catchments but intermediate catchments, serving as connections between other catchments. For example, they may correspond to only a very small river segment, making their size quite small.
- Answer for Q4: We have changed the caption of Fig. S1 to the following: "Figure S1. Distribution of the 2.6 million NHDPlus local catchment areas and flowline lengths."

**Comment**: The authors need to explain in the main text the format and size of all the files uploaded to Zenodo, especially detailing what information is included in the files with the format 'gpkg' and what software readers can use to open and edit them.

**Response**: We have added the detailed information (Lines 639-645) of all the files uploaded to Zenodo in the "Data and code availability" Section**

**Comment**: Among the 29,320 WQP stations, some stations do not have existing upstream watershed boundaries. In such cases, the authors obtained the watershed boundaries using DEM. I have the following questions regarding this:

1) The authors need to clarify, among the 22,201 stations, how many watershed boundaries were derived from DEM and how many from NHDPlus?

2) What resolution of DEM was used, and how were the watershed boundaries calculated?
3) The authors should compare their calculated watershed boundaries with the global watershed boundaries based on a 90-m resolution DEM and advanced algorithms (ESSD 6, 1151–1166, 2024) and present a comparison figure in the SI.

**Response**: Thanks for these clarification questions. Below are our answers: 1) All watershed boundaries of the 22,201 stations are derived using the HyRiver Python package (Chegini et al., 2021). This package does not directly derive catchment boundaries themselves but retrieves them through The Hydro Network-Linked Data Index (NLDI) web server. Additionally, the package simplifies catchment boundaries and splits the catchment at the location of the WQP stations.

2) We did not use DEM to obtain watershed boundaries. We apologize for the confusion due to lack of clarity earlier.

3) Since we did not calculate the boundaries ourselves and the WQP stations can be located at any position along rivers (not necessarily at the outlet), it is not comparable with the global watershed boundaries based on a 90-m resolution DEM and advanced algorithms (ESSD 6, 1151–1166, 2024). We have completely rewritten the corresponding paragraph (Lines 205-214) to eliminate any potential confusion.

**Comment**: The authors should provide a clear definition of "headwater" as used in this paper in the introduction. Is it determined based on stream order, river length, drainage area, or the number of tributaries? Additionally, they need to explain why they focus on headwaters.

**Response**: This has been addressed in our response to a previous comment. We will use the terminology "small catchment" hereafter, which refers to: a) DOC concentration at the outlet of a catchment is attributed to the entire upstream drainage area, and b) for small catchments, we can neglect the degradation of DOC in the stream at daily or sub-daily time steps, as explained in Section 2.1.

**Comment**: Maybe convert Table 2 into a bar chart and place it in SI. Additionally, again, many abbreviations in Table 2 need to be explained with their full terms.

**Response**: Table 2 has been converted into a bar chart (Figure 3), but we believe it should remain in the main text as it underscores the importance of the selected features and serves as a critical result for Section 3.1. Furthermore, the issue of abbreviations has been addressed in our earlier response by adding detailed explanations to the figure captions.

**Comment**: The language in this article needs further refinement. Here are just some examples that need to be revised, and the authors should check the entire text:

1) Delete "quickly" from line 149.

2) Delete "required for this study" from line 153.

3) Refine "We collect a wide range of environmental variables, comprising a total of 126 variables" to "We collect 126 environmental variables.".

4) Change "The ML technique used in this study is the eXtreme Gradient Boosting (XGBoost) algorithm" to "We use the eXtreme Gradient Boosting (XGBoost) ML algorithm."

**Response**: Thanks. We have carefully reviewed the entire text to identify and implement additional language refinements, enhancing overall clarity and readability. All the reviewer's specific comments have been accepted and incorporated into the main text.

**Comment**: The title is a bit long; it is recommended to change it to: "U.S. Transformation Rate Map of Dissolved Organic Carbon" or "Transformation Rate Map of Dissolved Organic Carbon in the Contiguous U.S."

**Response**: Great suggestion. We have changed the title to "Transformation Rate Maps of Dissolved Organic Carbon in the Contiguous U.S." to enhance clarity and conciseness

**Comment**: The citation format for figures is completely inconsistent throughout the text. Examples for the same figure include: Fig. S1, supplementary Fig. S1, and Supplementary Fig. S1. Please check the entire text (main text, figures, SI) and standardize according to ESSD requirements.

**Response**: We have thoroughly reviewed the entire text, including the main text, figures, and supplementary information, to ensure that all figure citations are standardized according to ESSD requirements. Specifically, we will consistently use the format "Supplementary Fig. S1" throughout the manuscript.

Comment: L175 ScienceBase also provides indicators of human activities, right?

**Response**: You are correct—ScienceBase provides catchment attributes across 11 categories, including human activities. In our study, we listed the four categories that were most significant in predicting DOC. We have refined the sentence on Lines 189–191 to clarify this point.

**Comment**: L244 "Out of the remaining 95 variables (see supplementary Tables S1 and S2 for details), 46 are relatively independent from each other. However, the other 49 are highly correlated with one or more variables." How did the authors determine "relatively independent" and "highly correlated"? I expect to see more explanation of this in the main text.

**Response**: Thanks for pointing this out. A Pearson correlation coefficient of  $\pm 0.8$  was used as the threshold to distinguish between "relatively independent" and "highly correlated" variables, following the guidelines outlined by Schober et al. (2018). We have updated the entire Section 2.2.3 to improve clarity and ensure better understanding.

**Comment**: Line 249, change "see Supplementary Figure S3" to "Supplementary Figure S3." Please check the entire text for similar instances where "see" is unnecessary.

**Response**: We have updated "see Supplementary Figure S3" to "Supplementary Fig. S3" on Line 249. Additionally, we have thoroughly reviewed the entire text to identify and remove any unnecessary instances of "see" in figure citations.

**Comment**: Line 251: "This new variable is thus independent of the other environmental variables." I do not understand the basis for this statement. Even if the new 9 combined parameters are formed, it is unlikely that they are completely independent of the other 46 parameters. The authors should provide a brief explanation in the main text or delete this sentence.

**Response**: We have opted to use the term "relatively independent" instead and have updated the entire Section 2.2.3 to enhance clarity and ensure better understanding.

Comment: Lines 273-275 need to be supported by references.

**Response**: The two sentences, "Recent studies have demonstrated the efficiency and effectiveness of these techniques in capturing high-dimensional and complex relationships between a target biogeochemical variable and various environmental predictors," and "These techniques have been successfully applied in various studies, including riverine sediment, beach water quality, oceanic particulate organic carbon, and eutrophication impacts from corn production (Abeshu et al., 2022; Li et al., 2022; Liu et al., 2021; Romeiko et al., 2020; Fan et al., 2021)," shared the same citations. We have revised these sentences for clarity and conciseness (Lines 293–297).

Comment: Line 379: "per\_canopy" is too difficult to understand.

**Response**: The term "per\_canopy" is no longer used; instead, the original NHDPlus name, "CNPY11\_BUFF100," is retained. When this variable appears in tables or figures, a short description, such as "areal percentage of canopy in the riparian buffer," will be provided in the corresponding footnotes or captions to ensure clarity.

**Comment**: In some places, it is written as "section," while in others, it is abbreviated as "sect" (e.g., L380).

**Response**: Thanks. We have reviewed the ESSD standards and have used "Sect." in running text, while using "Section" at the beginning of sentences, in accordance with the guidelines.

**Comment**: L413 'Note the unit of DOC concentration in water is mostly reported in mg/L (Schelker et al., 2012; Tian et al., 2015b; Langeveld et al., 2020)'. I think this sentence is not important to be in the main text.

Response: Deleted.

**Comment**: L481-482 "Blue, red, and grey colors are employed to indicate whether dropping the corresponding predictor will result in an increase, decrease, or insignificant change in the model's performance, respectively" should be in figure caption, rather than here.

**Response**: This sentence has been moved to the figure caption.

**Comments for dataset in Zenodo:**

**Comment**: There are many blank "nodata" areas within the CONUS\_DOC\_MAP, whereas the CONUS\_PR\_MAP does not have this issue. The authors need to explain this in the main text.

**Response**: It has already been explained in the first paragraph of section 5 as: "Due to missing data in the HWSD 1km SOC map at about 0.6 million NHDPlus local catchments, we cannot calculate the  $C_{DOC\ runoff}$  values over those catchments."

**Comment**: For reproducibility, the authors need to provide the shapefiles (or other similar vector data) for the 2595 watersheds used for machine learning training and the 3210 watersheds used for evaluation, as well as the shapefiles for these 5805 stations. The machine learning codes, as well as the raw data used for training the machine learning model, need to be uploaded to Zenodo; Then provide another link in the manuscript (not https://doi.org/10.5281/zenodo.8339372).

Response: Thank you. We have updated the Zenodo repository

(https://zenodo.org/records/14563816) to include the latest results as well as all necessary files required to reproduce our findings. Additionally, we have created a GitHub repository containing the code related to feature selection, model training, and evaluation. The repository is available at https://github.com/Ceyxleo/DOC-Param-Map.

**Suggestion for figures:**

**Comment**: The background color of all 2D density plots needs to be changed because the background color is included in the color scale. This makes it difficult for readers to distinguish between the data and the background color.

**Response**: Good catch! We have removed the background color of all 2D density plots (Figure 4, 7, S4, S7, S10 and S13).

**Comment**: Are the points in Figure 1 outlets or geometric centers of the watersheds? Additionally, it is necessary to indicate in the figure or caption that the gray lines represent rivers and the black lines represent national boundaries. Also, please specify the sources of these two elements.

**Response**: Thank you for the question. The points in the figure represent the locations of the WQP stations, which correspond to the outlets of their respective small catchments. The CONUS boundary was derived from the GeoPandas built-in shapefile data, accessible through geopandas.read\_file(gpd.datasets.get\_path('naturalearth\_lowres')). The river shapefile was sourced from Natural Earth. Since both datasets are open-source, we believe it is not essential to mention their sources in the main text. However, we have updated the caption of Figure 1 to clarify the meaning of the points and to include the data sources.

**Comment**: Figures 4 and 7 contain numerous abbreviations that are not explained in the captions, making it difficult for readers to understand the figures directly.

Response: This issue has been addressed previously by adding explanations to the captions.

**Comment**: It is necessary to explain in the caption of Figure 4 what the correlation coefficient is. Is it Spearman rank?

**Response**: The sentence "The Pearson correlation coefficient is used" has been added to the caption of Figure 5 (previously Figure 4) for improved clarity.

Comment: Why are there many nodata areas near the national boundaries in Figure 5?

**Response**: Nice catch! This discrepancy is not due to missing data but rather a mismatch between two geodatasets. The country boundaries, obtained from gpd.datasets.get\_path("naturalearth\_lowres") in the GeoPandas library, are of very low resolution. In contrast, the NHDPlus local catchments are derived from a 30m DEM, which are much more accurate. This resolution difference leads to discrepancies at the national boundaries. Additionally, for approximately 14,000 NHDPlus local catchments, we cannot retrieve their catchment attributes, so they are removed from the prediction set.

Comment: Fig. S1 needs ticks on the X-axis.

Response: added.

**Comment**: In the main manuscript, I do not understand the differences between the two types of watershed boundaries provided by NHDPlus. Besides, I do not understand Figure S2. It is recommended to use a real terrain example for illustration to show the differences between these two kinds of watershed boundaries. For example, based on Google Earth, mark the river, the two different watersheds, and the DOC station location (outlet).

**Response**: Upon rechecking the NHDPlus Version 2.1 National Seamless Geodatabase (.gdb), we discovered that it contains only the boundaries of local catchments. We apologize for the earlier inaccurate information and have updated the corresponding paragraphs (Lines 205–214 and Lines 229–237) in the main text accordingly. To enhance clarity, we have also regenerated Supplementary Figure S2 using real catchment boundaries. The updated figure includes two subplots: a) illustrating the relationship between a small catchment and the NHDPlus local catchments it contains, and b) showing the nesting of small catchments and identifying those used for model training and validation.

**Comment**: I don't have research experience with DOC; most of my comments are from a geomorphological perspective, as well as regarding readability and clarity. I hope my suggestions are helpful.

**Response**: Yes, your suggestions are indeed very helpful for improving the readability and clarity. We very much appreciate them.

Reviewer #2:

The paper titled "Deriving a Transformation Rate Map of Dissolved Organic Carbon over the Contiguous U.S." presents a novel formula that directly links soil organic carbon (SOC) concentration with dissolved organic carbon (DOC) concentration in headwater streams. This formula uses a single parameter, the transformation rate (Pr), to represent the overall processes governing the conversion of SOC to DOC and its leaching from soils into headwater streams (along with runoff). The authors have developed a high-resolution Pr map for the contiguous U.S. (CONUS), which is robustly derived and empirically validated. This map provides a crucial foundation for improving the simulation of the terrestrial carbon cycle in land surface and Earth system models.

The study is well-organized and well-written, demonstrating high novelty and significantly contributing to riverine DOC modeling. The paper outlines several potential applications of the derived products. Additionally, the authors have thoroughly addressed the uncertainty analysis and limitations of the study. I recommend accepting the manuscript in its current form

**Response**: We greatly appreciate the positive comments from Reviewer #2.

**Reviewer #3:**

The manuscript addresses a significant issue in the field of environmental science, particularly in understanding the dynamics of dissolved organic carbon (DOC) in the context of climate change and carbon cycling. The manuscript makes effective use of the public available datasets, especially the Water Quality Portal (WQP), provide a solid foundation for the analysis. The proposed method of estimating transformation rates from soil organic carbon (SOC) to DOC using a lumped parameter approach is innovative and could simplify large-scale modeling efforts. The model's simplicity and the reduced data requirements are strengths, making it more accessible for application in regions with limited data availability. And lastly, the model's potential to predict riverine DOC concentrations from SOC values is a valuable tool for water quality management and environmental monitoring. However, there are some potential weaknesses for the authors to consider and to improve the quality of the manuscript: (1) Generalizability: The study focuses on the contiguous U.S., and it is unclear how well the findings and models could be generalized to other regions with different environmental conditions. (2) Complexity of DOC Dynamics: The simplification of the model might overlook the complexity of DOC dynamics, including the influence of various biotic and abiotic factors. (3) Validation and Calibration: The manuscript would benefit from a more detailed discussion on the validation and calibration of the model, including the use of independent datasets. (4) Potential Over-simplification: The assumption that riverine DOC degradation in headwater streams is negligible might be an oversimplification, especially in ecosystems with high microbial activity. (5) Lack of Experimental Data: The study relies heavily on existing datasets, and there is a lack of experimental data to support the model's predictions. Overall, the development of a predictive model that can estimate riverine DOC concentrations from SOC values is innovative and has practical applications, I would recommend the manuscript for acceptance with major revision.

**Response**: We appreciate the reviewer's insightful comments. In fact, many of these points were central during the planning and implementation phases of this study. It is common for modelers to face a dilemma between complexity and simplicity. According to the principle of Occam's Razor (Walsh, 1979), complexity does not always bring better model predictive power, particularly in cases where the scientific community doesn't yet have a clear understanding of the relevant processes. In our case, it is our observation that the land modeling and biogeochemical science communities have not yet achieved a clear understanding of the DOC dynamics, as evidenced by the diverse descriptions of DOC leaching processes in existing models, such as DLEM, INCA-C, JULESDOCM, ECO3D, and TRIPLEX-HYD. More specifically, the communities are still unclear about 1) how many specific processes are involved from the land to aquatic ecosystems regarding DOC dynamics, 2) whether our understanding of each specific process is clear enough to allow robust mathematical formulations (aka, governing equations), and 3) how we can parameterize each governing equation to effectively account for spatiotemporal heterogeneities in the relevant controlling factors. It is also our observation that the currently available observations (known variables) are too limited compared to the number of parameters in existing models (unknown variables) to enable parsimonious process descriptions, i.e., overparameterization. For instance, for modeling riverine DOC at the regional and larger scales, to the best of our knowledge, the only observation data available at the corresponding scales are the DOC observations from the river gauges. Based on these rationales, we proposed

our simplified formula with the hope that it is complementary to the existing, pioneering modeling approaches. As the reviewer rightfully pointed out, this "lumped parameter approach" has the advantages of "simplicity and the reduced data requirements", allowing for the usage of machine-learning techniques in the parameterization strategy. More importantly, the resulting parameter map is indeed effective, as demonstrated in our other ongoing modeling study, where we used the parameter map as a key input to a land-river modeling framework for DOC, validated the model simulated riverine DOC concentration values against the observed at over 450 large river gauges over the contiguous U.S., where the drainage areas of the gauges range from 55 km2 to  $1.1 \times 10^6$  km2. Our modeling results are still preliminary since we are still adding and debugging the coding of other relevant processes, but both R-square and Kling-Gupta efficiency exceed 0.6 already, suggesting the fidelity of the parameterization strategy in the context of regional-scale DOC modeling. That said, given that our current study is mainly a dataset development effort tailored for ESSD, not a full-scale modeling one, we will report our modeling study in a separate manuscript.

**Figure R1**: Spatial distribution of DOC over 450 river gauges with DOC observations (top) and comparison between the simulated and observed long-term average riverine DOC concentrations at these stations (bottom). *Note: We provide these figures here only as part of our responses to*

**the reviewer's comments. It is NOT our intention to publish these figures as part of the manuscript under review here.**

Next we provide point-to-point responses to each major comment:

- 1) We believe that our lumped parameter approach and machine learning-based parameterization strategy are generalizable to other regions. The conceptualization of the lumped parameter approach itself is generic and not site-specific. The machine learning techniques are not site-specific either. Moreover, the contiguous U.S. as a study area itself contains significant spatial heterogeneities of environmental conditions, including diverse vegetation types, soil compositions, topographic variations, and climate regimes, that can be found elsewhere. In fact, one of our next steps is to expand our methodology framework over the global domain and produce a global parameter map, which will be reported in a separate study. At the global scale, the data availability is understandably less than in the U.S. Our tentative plan to overcome this limitation includes but not limited to: 1) collect as much observational data as possible, particularly riverine DOC observations, from public datasets and literature taking advantage of modern AI techniques; 2) call for more field work to collect DOC observations; 3) caution the unavoidably larger uncertainties embedded in the global parameter map (compared to the U.S. map). We have added some discussions about the generalizability of our ML-based parameterization into Lines 626-636 in the Section 5.
- 2) We fully acknowledge the complexity of DOC dynamics, particularly concerning lateral leaching processes in soil. To address this, we have added a discussion on the impact of biotic and abiotic factors in the Introduction section (Lines 54–57). While existing process-based DOC models do not comprehensively capture all these mechanisms, they also employ various simplified representations (Lines 70–76), which, as we previously discussed, require extensive parameterization (Lines 78–80). Therefore, we propose our study as a first step towards a new pathway to advance the understanding of DOC dynamics that is complementary to the existing modeling approaches.
- 3) We agree that it is important to have independent datasets for calibration and validation. It appears that we may not have provided a clear description about the validation dataset. Our validation catchments are NOT within but encompass the catchments we used for the ML modeling. Therefore, the validation strategy we applied is appropriate for a dataset study. To enhance clarity, we have updated the corresponding paragraph (Lines 229–237) in Section 2.2.3 and regenerated Supplementary Figure S2b using real catchment boundaries. The updated figure and text illustrate the nesting of small catchments and clarifies how independent catchments used for model training and evaluation catchments used for model validation are selected.
- 4) We respectfully argue that our assumption is valid for most, if not all, headwater streams. There are two rationales behind our assumption: 1) Based on a literature review, we summarized the DOC degradation rates used in existing process-based modeling studies and reported by the experimental studies, as listed in Table S1. All of these studies suggest that, for headwater streams, the in-stream DOC degradation rate is approximately 0.01 per day; 2) Typical residence time of DOC in headwater streams (from the moment it enters into streams from soils to the moment it leaves the headwater streams into downstream rivers) is on the order of a couple of hours, i.e., much less than a day

(Ducharne et al., 2003; Li et al., 2013). Taken together, it is reasonable to assume that the DOC degradation is negligible between the moment it enters the streams from soils and the moment it leaves the headwater streams, hence supporting Eqn. (5).

5) On one hand, we consider observed riverine DOC data to be a highly reliable source for validation. On the other hand, since there are no direct measurements of Pr, we suggest that new field experiments could be designed and implemented based on our lumped parameter approach. To address this concern, we have added a few sentences (Lines 621-623) in Section 5.

**In addition, I have a few minor comments; please see below:**

**Response**: These minor comments are quite helpful as well. We provide our point-to-point responses to them, in blue color, in the following.

**Comment**: Line 131: "Eqn. (4) has several advantages" change to "Eqn. (4) has two advantages".

**Response: Changed.**

**Comment**: Line 153: There are much higher spatial resolution SOC data available (e.g. SoilGrids provides 250m resolution data available, see reference below), why chose use HWSD?

Hengl, Tomislav, Jorge Mendes de Jesus, Gerard BM Heuvelink, Maria Ruiperez Gonzalez, Milan Kilibarda, Aleksandar Blagotić, Wei Shangguan et al. "SoilGrids250m: Global gridded soil information based on machine learning." PLoS one 12, no. 2 (2017): e0169748.

**Response**: Thank you for highlighting this point. In designing this project, we selected the HWSD dataset as it was one of the very first globally harmonized soil datasets, integrating data from diverse national and regional sources into a standardized framework. This makes it a foundational resource for many Earth system modeling studies (Best et al., 2011; Zhao et al., 2018). Additionally, as discussed previously, the Pr map derived serves as a critical input for our subsequent Earth system model development study, the land module of which relies on HWSD data. Upon reviewing the SoilGrids data, we recognize its value as a robust and increasingly popular soil data source for Earth system modeling studies. Therefore, we have applied the same methodology to generate a Pr map using the latest SoilGrids2.0 dataset (Poggio et al., 2021). A discussion of both SOC datasets has been added in Section 2.2 (Lines 165–178). Given the similar results and discussions, the HWSD-based model remains the primary focus of the main text. Supplementary materials (Supplementary Table S4–S6 and Supplementary Figures S8–S14) include additional information on the SoilGrids-based model. Users can select the Pr map that best suits their specific needs.

**Comment**: Line 219-220: According to the description, 3210 pairs for evaluation are within the catchment of 2595 pairs for ML modeling, therefore they are not independent and the evaluation might biased.

**Response**: The evaluation catchments are not within the independent catchments, but rather, they encompass the independent catchments. In cases of paired & nested catchments, we take the one with a smaller drainage area for developing our ML model, and leave the one with a larger

drainage area for future validation. Hence, our validation strategy is effective. To avoid further confusion, we have updated Supplementary Figure S2 to reflect the actual boundaries of nested catchments and revised the text (Lines 229–237) accordingly.

**Comment**: Figure 3: In scatter plots, observed data are typically placed on the y-axis, while simulated data are positioned on the x-axis. I suggest moving the estimated Pr to the y-axis and the simulated Pr to the x-axis. The same recommendation applies to Figure 6.

**Response**: We have looked into several recently published articles in ESSD and found that observed data are mostly placed on the horizontal x-axis instead. Therefore, we will respectfully keep our current axis arrangement, following the more conventional approach.

**Reference:**

Best, M. J., Pryor, M., Clark, D. B., Rooney, G. G., Essery, R. . L. H., Ménard, C. B., Edwards, J. M., Hendry, M. A., Porson, A., Gedney, N., Mercado, L. M., Sitch, S., Blyth, E., Boucher, O., Cox, P. M., Grimmond, C. S. B., and Harding, R. J.: The Joint UK Land Environment Simulator (JULES), model description – Part 1: Energy and water fluxes, Geosci Model Dev, 4, 677–699, https://doi.org/10.5194/gmd-4-677-2011, 2011.

Chegini, T., Li, H.-Y., and Leung, L.: HyRiver: Hydroclimate Data Retriever, J Open Source Softw, 6, 3175, https://doi.org/10.21105/joss.03175, 2021.

Ducharne, A., Golaz, C., Leblois, E., Laval, K., Polcher, J., Ledoux, E., and De Marsily, G.: Development of a high resolution runoff routing model, calibration and application to assess runoff from the LMD GCM, J Hydrol (Amst), 280, 207–228, https://doi.org/10.1016/S0022-1694(03)00230-0, 2003.

Li, H., Wigmosta, M. S., Wu, H., Huang, M., Ke, Y., Coleman, A. M., and Leung, L. R.: A physically based runoff routing model for land surface and earth system models, J Hydrometeorol, 14, 808–828, https://doi.org/10.1175/JHM-D-12-015.1, 2013.

Poggio, L., De Sousa, L. M., Batjes, N. H., Heuvelink, G. B. M., Kempen, B., Ribeiro, E., and Rossiter, D.: SoilGrids 2.0: Producing soil information for the globe with quantified spatial uncertainty, SOIL, 7, 217–240, https://doi.org/10.5194/soil-7-217-2021, 2021.

Schober, P., Boer, C., and Schwarte, L. A.: Correlation Coefficients: Appropriate Use and Interpretation, Anesth Analg, 126, 1763–1768, https://doi.org/10.1213/ANE.00000000002864, 2018.

Walsh, D.: Occam's razor: A principle of intellectual elegance, American Philosophical Quarterly, 16, 241–244, 1979.

Zhao, M., Golaz, J. C., Held, I. M., Guo, H., Balaji, V., Benson, R., Chen, J. H., Chen, X., Donner,
L. J., Dunne, J. P., Dunne, K., Durachta, J., Fan, S. M., Freidenreich, S. M., Garner, S. T., Ginoux,
P., Harris, L. M., Horowitz, L. W., Krasting, J. P., Langenhorst, A. R., Liang, Z., Lin, P., Lin, S.
J., Malyshev, S. L., Mason, E., Milly, P. C. D., Ming, Y., Naik, V., Paulot, F., Paynter, D.,
Phillipps, P., Radhakrishnan, A., Ramaswamy, V., Robinson, T., Schwarzkopf, D., Seman, C. J.,
Shevliakova, E., Shen, Z., Shin, H., Silvers, L. G., Wilson, J. R., Winton, M., Wittenberg, A. T.,
Wyman, B., and Xiang, B.: The GFDL Global Atmosphere and Land Model AM4.0/LM4.0: 2.

Model Description, Sensitivity Studies, and Tuning Strategies, J Adv Model Earth Syst, 10, 735–769, https://doi.org/10.1002/2017MS001209, 2018.

---

## Author Response (AR2)

In the following, the reviewer's comments, our point-to-point responses, our revisions are shown in black, blue, and purple colors, respectively.

Editor:

Reviewers are in general satisfied with the revisions and recommand for publication. Before taking this decision, I suggest for minor revision with following comments (noting that all line numbers refer to the tracked-changes version of the revised manuscript).

Response: Thank you for your careful review of our revised manuscript and for providing constructive feedback. We appreciate the reviewers' overall satisfaction with the revisions and their recommendation for publication. We have addressed the minor comments as outlined below.

Comment: Line 14: In the revision, for "earth system models," why is only the first letter of "Earth" capitalized? Shouldn't all three words be capitalized?

Response: We have corrected the capitalization to "Earth System Models" and "Earth System Modeling" throughout the text to ensure consistency with standard scientific terminology. All three words are now capitalized.

Comment: Line 57: ", pH" changed to ", and pH".

Response: Changed.

Comment: Line 280: Change "Figure S8" to "Supplementary Fig. S8" to keep the style consistent in the entire manuscript. Another important consideration is that all figures, including supplementary figures, should be cited in sequential order within the main text, which is currently not the case. For instance, while Supplementary Fig. S8 is referenced here, Supplementary Figs. S3 through S7 have not been mentioned or cited in the preceding paragraph. All the figures and tables should be checked to avoid this issue.

Response: We have updated the figure citation format to "Supplementary Fig. Sxx". Additionally, we have carefully reviewed the entire manuscript and Supplementary to ensure that all figures (including subplots) and tables, are cited in sequential order within the main text.